# FEDBABU: TOWARD ENHANCED REPRESENTATION FOR FEDERATED IMAGE CLASSIFICATION

**Jaehoon Oh**[*]
Graduate School of KSE, KAIST
jhoon.oh@kaist.ac.kr

**Sangmook Kim,**[*] **Se-Young Yun**
Graduate School of AI, KAIST
{sangmook.kim, yunseyoung}@kaist.ac.kr

## ABSTRACT

Federated learning has evolved to improve a single global model under data heterogeneity (as a curse) or to develop multiple personalized models using data heterogeneity (as a blessing). However, little research has considered both directions simultaneously. In this paper, we first investigate the relationship between them by analyzing Federated Averaging (McMahan et al., 2017) at the client level and determine that a better federated global model performance does not constantly improve personalization. To elucidate the cause of this personalization performance degradation problem, we decompose the entire network into the body (extractor), which is related to universality, and the head (classifier), which is related to personalization. We then point out that this problem stems from training the head. Based on this observation, we propose a novel federated learning algorithm, coined FedBABU, which only updates the body of the model during federated training (i.e., the head is randomly initialized and *never* updated), and the head is fine-tuned for personalization during the evaluation process. Extensive experiments show consistent performance improvements and an efficient personalization of FedBABU. The code is available at https://github.com/jhoon-oh/FedBABU.

## 1 INTRODUCTION

Federated learning (FL) (McMahan et al., 2017), which is a distributed learning framework with personalized data, has become an attractive field of research. From the early days of this field, improving *a single global model* across devices has been the main objective (Zhao et al., 2018; Duan et al., 2019; Li et al., 2018; Acar et al., 2021), where the global model suffers from data heterogeneity. Many researchers have recently focused on *multiple personalized models* by leveraging data heterogeneity across devices as a blessing in disguise (Chen et al., 2018; Dinh et al., 2021; Zhang et al., 2020; Fallah et al., 2020; Shamsian et al., 2021; Smith et al., 2017). Although many studies have been conducted on each research line, a lack of research remains on how to train a good global model for personalization purposes (Ji et al., 2021; Jiang et al., 2019). In this study, for personalized training, each local client model starts from a global model that learns information from all clients and leverages the global model to fit its own data distribution.

Jiang et al. (2019) analyzed personalization methods that adapt to the global model through fine-tuning on each device. They observed that the effects of fine-tuning are encouraging and that training in a central location increases the initial accuracy (of a single global model) while decreasing the personalized accuracy (of on-device fine-tuned models). We focus on *why* opposite changes in the two performances appear. This suggests that the factors for universality and personalization must be dealt with separately, inspiring us to decouple the entire network into the body (i.e., extractor) related to generality and the head (i.e., classifier) related to specialty, as in previous studies (Kang et al., 2019; Yu et al., 2020; Yosinski et al., 2014; Devlin et al., 2019) for advanced analysis. Kang et al. (2019) and Yu et al. (2020) demonstrated that the head is biased in class-imbalanced environments. Note that popular networks such as MobileNet (Howard et al., 2017) and ResNet (He et al., 2016) have only one linear layer at the end of the model, and the head is defined as this linear layer whereas the body is defined as all of the layers except the head. The body of the model is related to representation learning and the head of the model is related to linear decision boundary learning. We shed light on the cause of the personalization performance degradation by decoupling parameters, pointing out that such a problem stems from training the head.

---
[*]Equal Contribution

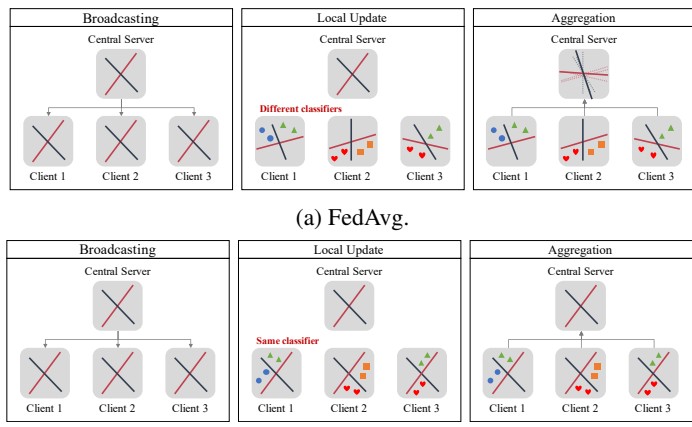

(a) FedAvg.

(b) FedBABU.

Figure 1: **Difference in the local update and aggregation stages between FedAvg and FedBABU**. In the figure, the lines represent the decision boundaries defined by the head (i.e., the last linear classifier) of the network; different shapes indicate different classes. It is assumed that each client has two classes. (a) FedAvg updates the entire network during local updates on each client and then the local networks are aggregated entirely. Therefore, the heads of all clients and the head of the server are different. Whereas, (b) FedBABU only updates the body (i.e., all of the layers except the head) during local updates on each client and then the local networks are aggregated body-partially. Therefore, the heads of all clients and the head of the server are the same.

Inspired by the above observations, we propose an algorithm to learn a single global model that can be efficiently personalized by simply changing the Federated Averaging (FedAvg) (McMahan et al., 2017) algorithm. FedAvg consists of four stages: client sampling, broadcasting, local update, and aggregation. In the client sampling stage, clients are sampled because the number of clients is so large that not all clients can participate in each round. In the broadcasting stage, the server sends a global model (i.e., an initial random model at the first broadcast stage or an aggregated model afterward) to participating clients. In the local update stage, the broadcast model of each device is trained based on its own data set. In the aggregation stage, locally updated models are sent to the server and are aggregated by averaging. Among the four stages, we focus on the *local update* stage from both the universality and personalization perspectives. *Here, we only update the body of the model in the local update stage, i.e., the head is never updated during federated training.* From this, we propose FedBABU, **Fed**erated Averaging with **B**ody **A**ggregation and **B**ody **U**pdate. Figure 1 describes the difference during the local update and aggregation stages between FedAvg and FedBABU. *Intuitively, the fixed head can be interpreted as the criteria or guideline for each class and our approach is representation learning based on the same fixed criteria across all clients during federated training.* This simple change improves the representation power of a single global model and enables the more accurate and rapid personalization of the trained global model than FedAvg.

Our contributions are summarized as follows:

- We investigate the connection between a single global model and fine-tuned personalized models by analyzing the FedAvg algorithm at the client level and show that training the head using shared data on the server negatively impacts personalization.
- We demonstrate that the fixed random head can have comparable performance to the learned head in the centralized setting. From this observation, we suggest that sharing the fixed random head across all clients can be more potent than averaging each learned head in federated settings.
- We propose a novel algorithm, **FedBABU**, that reduces the update and aggregation parts from the entire model to the body of the model during federated training. We show that FedBABU is efficient, particularly under more significant data heterogeneity. Furthermore, a single global model trained with the FedBABU algorithm can be personalized rapidly (even with one fine-tuning epoch). We observe that FedAvg outperforms most of preexisting personalization FL algorithms and FedBABU outperforms FedAvg in various FL settings.
- We adapt the body update and body aggregation idea to the regularization-based global federated learning algorithm (such as FedProx (Li et al., 2018)). We show that regularization reduces the personalization capabilities and that this problem is mitigated through BABU.

## 2    RELATED WORKS

**FL for a Single Global Model** Canonical federated learning, FedAvg proposed by McMahan et al. (2017), aims to learn a single global model that collects the benefits of affluent data without storing the clients' raw data in a central server, thus reducing the communication costs through local updates. However, it is difficult to develop a globally optimal model for non-independent and identically distributed (non-IID) data derived from various clients. To solve this problem, studies have been conducted that make the data distribution of the clients IID-like or add regularization to the distance from the global model during local updates. Zhao et al. (2018) suggested that all clients share a subset of public data, and Duan et al. (2019) augmented data to balance the label distribution of clients. Recently, two studies (Li et al., 2018; Acar et al., 2021) penalized local models that have a large divergence from the global model, adding a regularization term to the local optimization process and allowing the global model to converge more reliably. However, it should be noted that the single global model trained using the above methods is not optimized for each client.

**Personalized FL** Personalized federated learning aims to learn personalized local models that are stylized to each client. Although local models can be developed without federation, this method suffers from data limitations. Therefore, to maintain the benefits of the federation and personalized models, many other methods have been applied to FL: clustering, multi-task learning, transfer learning, regularized loss function, and meta-learning. Two studies (Briggs et al., 2020; Mansour et al., 2020) clustered similar clients to match the data distribution within a cluster and learned separate models for each cluster without inter-cluster federation. Similar to clustering, multi-task learning aims to learn models for related clients simultaneously. Note that clustering ties related clients into a single cluster, whereas multi-task learning does not. The generalization of each model can be improved by sharing representations between related clients. Smith et al. (2017) and Dinh et al. (2021) showed that multi-task learning is an appropriate learning scheme for personalized FL. Transfer learning is also a recommended learning scheme because it aims to deliver knowledge among clients. In addition, Yang et al. (2020) and Chen et al. (2020b) utilized transfer learning to enhance local models by transferring knowledge between related clients. T Dinh et al. (2020) and Li et al. (2021) added a regularizer toward the global or average personalized model to prevent clients' models from simply overfitting their own dataset. Unlike the aforementioned methods that developed local models during training, Chen et al. (2018) and Fallah et al. (2020) attempted to develop a good initialized shared global model using bi-level optimization through a Model-Agnostic Meta-Learning (MAML) approach (Finn et al., 2017). A well-initialized model can be personalized through updates on each client (such as inner updates in MAML). Jiang et al. (2019) argued that the FedAvg algorithm could be interpreted as a meta-learning algorithm, and a personalized local model with high accuracy can be obtained through fine-tuning from a global model learned using FedAvg. In addition, various technologies and algorithms for personalized FL have been presented and will be helpful to read; see Tan et al. (2021) and Kulkarni et al. (2020) for more details.

**Decoupling the Body and the Head for Personalized FL** The training scheme that involves decoupling the entire network into the body and the head has been used in various fields, including long-tail recognition (Kang et al., 2019; Yu et al., 2020), noisy label learning (Zhang & Yao, 2020), and meta-learning (Oh et al., 2021; Raghu et al., 2019).[1] For personalized FL, there have been attempts to use this decoupling approach. For a consistent explanation, we describe each algorithm from the perspective of local update and aggregation parts. FedPer (Arivazhagan et al., 2019), similar to FedPav (Zhuang et al., 2020), learns the entire network jointly during local updates and only aggregates the bottom layers. When the bottom layers are matched with the body, the body is shared on all clients and the head is personalized to each client. LG-FedAvg (Liang et al., 2020) learns the entire network jointly during local updates and only aggregates the top layers based on the pre-trained global network via FedAvg. When the top layers are matched with the head, the body is personalized to each client and the head is shared on all clients. FedRep (Collins et al., 2021) learns the entire network sequentially during local updates and only aggregates the body. In the local update stage, each client first learns the head only with the aggregated representation and then learns the body only with its own head within a single epoch. Unlike the above decoupling methods, we propose a FedBABU algorithm that learns only the body with the randomly initialized head during local updates and aggregates only the body. It is thought that fixing the head during the entire federated training provides the same guidelines on learning representations across all clients. Then, personalized local models can be obtained by fine-tuning the head.

---

[1]Although there are more studies related to decoupling parameters (Rusu et al., 2018; Lee & Choi, 2018; Flennerhag et al., 2019; Chen et al., 2019b), we focus on decoupling the entire network into the body and head.

## 3 PRELIMINARIES

**FL training procedure** We summarize the training procedure of FL following the aforementioned four stages with formal notations. Let $\{1, \cdots, N\}$ be the set of all clients. Then, the participating clients in the communication round $k$ with client fraction ratio $f$ is $C^k = \{C_i^k\}_{i=1}^{\lfloor Nf \rfloor}$. By broadcasting, the local parameters of the participating clients $\{\theta_i^k(0)\}_{i=1}^{\lfloor Nf \rfloor}$ are initialized by the global parameter $\theta_G^{k-1}$, that is, $\theta_i^k(0) \leftarrow \theta_G^{k-1}$ for all $i \in [1, \lfloor Nf \rfloor]$ and $k \in [1, \mathrm{K}]$. Note that $\theta_G^0$ is randomly initialized first. On its own device, each local model is updated using a locally kept data set. After local epochs $\tau$, the locally updated models become $\{\theta_i^k(\tau I_i^k)\}_{i=1}^{\lfloor Nf \rfloor}$, where $I_i^k$ is the number of iterations of one epoch on client $C_i^k$ (i.e., $\lceil \frac{n_{C_i^k}}{B} \rceil$), $n_{C_i^k}$ is the number of data samples for client $C_i^k$, and $B$ is the batch size. Therefore, $\tau I_i^k$ is the total number of updates in one communication interval. Note that our research mainly deals with a balanced environment in which all clients have the same size data set (i.e., $I_i^k$ is a constant for all $k$ and $i$).[2] Finally, the global parameter $\theta_G^k$ is aggregated by $\sum_{i=1}^{\lfloor Nf \rfloor} \frac{n_{C_i^k}}{n} \theta_i^k(\tau I_i^k)$, where $n = \sum_{i=1}^{\lfloor Nf \rfloor} n_{C_i^k}$, at the server. For our algorithm, the parameters $\theta$ are decoupled into the body (extractor) parameters $\theta_{ext}$ and the head (classifier) parameters $\theta_{cls} \in \mathbb{R}^{C \times d}$, where $d$ is the dimension of the last representations and $C$ is the number of classes.

**Experimental setup** We mainly use MobileNet on CIFAR100.[3] We set the number of clients to 100 and then each client has 500 training data and 100 test data; the classes in the training and test data sets are the same. For the heterogeneous distribution of client data, we refer to the experimental setting in McMahan et al. (2017). We sort the data by labels and divide the data into the same-sized shards. Because there is no overlapping data between shards, the size of a shard is defined by $\frac{|D|}{N \times s}$, where $|D|$ is the data set size, $N$ is the total number of clients, and $s$ is the number of shards per user. We control FL environments with three hyperparameters: client fraction ratio $f$, local epochs $\tau$, and shards per user $s$. $f$ is the number of participating clients out of the total number of clients in every round and a small $f$ is natural in the FL settings because the total number of clients is numerous. The local epochs $\tau$ are equal to the interval between two consecutive communication rounds. To fix the number of total updates to ensure the consistency in all experiments, we fix the product of communication rounds and local epochs to 320 (e.g., if local epochs are four, then the total number of communication rounds is 80). The learning rate starts with 0.1 and is decayed by a factor of 0.1 at half and three-quarters of total updates. $\tau$ is closely related to the trade-off between accuracy and communication costs. A small $\tau$ provides an accurate federation but requires considerable communication costs. $s$ is related to the maximum number of classes each user can have; hence, as $s$ decreases, the degree of data heterogeneity increases.

**Evaluation** We calculate the *initial accuracy* and *personalized accuracy* of FedAvg and FedBABU following the federated personalization evaluation procedure proposed in Wang et al. (2019) to analyze the algorithms at the client level: (1) the learned global model is broadcast to all clients and is then evaluated on the test data set of each client $D_i^{ts}$ (referred to as the *initial accuracy*), (2) the learned global model is personalized using the training data set of each client $D_i^{tr}$ by fine-tuning with the fine-tuning epochs of $\tau_f$; the personalized models are then evaluated on the test data set of each client $D_i^{ts}$ (referred to as the *personalized accuracy*). In addition, we calculate the *personalized accuracy* of other personalized FL algorithms (such as FedPer, LG-FedAvg, and FedRep). Algorithm 2 in Appendix A describes the evaluation procedure. The values (X±Y) in all tables indicate the mean±standard deviation of the accuracies *across all clients*, not across multiple seeds. Here, reducing the variance over the clients could be interesting but goes beyond the scope of this study.

## 4 PERSONALIZATION OF A SINGLE GLOBAL MODEL

We first investigate personalization of a single global model using the FedAvg algorithm, following Wang et al. (2019) and Jiang et al. (2019), to connect a single global model to multiple personalized models. We evaluate both the initial accuracy and the personalized accuracy, assuming that the test data sets are not gathered in the server but scattered on the clients.

---

[2] Appendix H reports the results under unbalanced and non-IID derived Dirichlet distribution.

[3] We also use 3convNet on EMNIST, 4convNet on CIFAR10, and ResNet on CIFAR100. Appendix I, F, and G report results, respectively. Appendix A presents the details of the architectures used.

Table 1: Initial and personalized accuracy of FedAvg on CIFAR100 under various FL settings with 100 clients. MobileNet is used. The initial and personalized accuracy indicate the evaluated performance without fine-tuning and after five fine-tuning epochs for each client, respectively.

| FL settings | | $s$=100 (heterogeneity ↓) | | $s$=50 | | $s$=10 (heterogeneity ↑) | |
|---|---|---|---|---|---|---|---|
| $f$ | $\tau$ | Initial | Personalized | Initial | Personalized | Initial | Personalized |
| | 1 | 46.93±5.47 | 51.93±5.19 | 45.68±5.50 | 57.84±5.08 | 37.27±6.97 | 77.46±5.78 |
| 1.0 | 4 | 37.44±4.98 | 42.66±5.09 | 36.05±4.04 | 47.17±4.26 | 24.17±5.50 | 70.41±6.83 |
| | 10 | 29.58±4.87 | 34.62±4.97 | 29.57±4.29 | 40.59±5.23 | 17.85±7.38 | 63.51±7.38 |
| | 1 | 39.07±5.22 | 43.92±5.55 | 38.20±5.73 | 49.55±5.36 | 29.12±7.11 | 71.24±7.82 |
| 0.1 | 4 | 35.39±4.58 | 39.67±5.21 | 33.49±4.72 | 43.63±4.77 | 21.14±6.86 | 67.14±6.72 |
| | 10 | 28.18±4.83 | 33.13±5.22 | 27.34±4.96 | 38.09±5.17 | 14.40±5.64 | 62.67±6.52 |

Table 2: Initial and personalized accuracy of FedAvg on CIFAR100 under a realistic FL setting ($N$=100, $f$=0.1, $\tau$=10) according to $p$, which is the percentage of all client data that the server also has. Here, the entire (or, full) network or body is updated on the server using the available data.

| $p$ | $s$=100 (heterogeneity ↓) | | $s$=50 | | $s$=10 (heterogeneity ↑) | |
|---|---|---|---|---|---|---|
| | Initial | Personalized | Initial | Personalized | Initial | Personalized |
| 0.00 | 28.18±4.83 | 33.13±5.22 | 27.34±4.96 | 38.09±5.17 | 14.40±5.64 | 62.67±6.52 |
| 0.05 (Full) | 29.23±5.03 | 32.59±5.24 | 27.13±4.34 | 34.34±4.78 | 18.22±5.64 | 54.68±6.77 |
| 0.05 (Body) | 28.50±4.93 | 33.03±5.36 | 27.96±4.86 | 39.10±5.55 | 14.78±5.59 | 60.19±6.46 |
| 0.10 (Full) | 30.59±4.93 | 33.34±5.30 | 29.62±4.27 | 35.50±4.84 | 19.24±5.15 | 49.62±7.48 |
| 0.10 (Body) | 32.90±4.77 | 36.82±4.66 | 32.81±4.97 | 40.80±5.62 | 18.35±6.75 | 60.94±7.30 |

Table 1 describes the accuracy of FedAvg on CIFAR100 according to different FL settings ($f$, $\tau$, and $s$) with 100 clients. The initial and personalized accuracies indicate the evaluated performance without fine-tuning and with five fine-tuning epochs for each client, respectively. As previous studies have shown, the more realistic the setting (i.e., a smaller $f$, larger $\tau$, and smaller $s$), the lower the initial accuracy. Moreover, the tendency of the personalized models to converge higher than the global model observed in Jiang et al. (2019) is the same. More interestingly, it is shown that the gap between the initial accuracy and the personalized accuracy increases significantly as the data become more heterogeneous. It is thought that a small $s$ makes local tasks easier because the label distribution owned by each client is limited and the number of samples per class increases.

In addition, we conduct an intriguing experiment in which the initial accuracy increases but the personalized accuracy decreases, maintaining the FL training procedures. In Jiang et al. (2019), the authors showed that centralized trained models are more difficult to personalize. Similarly, we design an experiment in which the federated trained models are more difficult to personalize. We assume that the server has a small portion of $p$ of the non-private client data of the clients[4] such that the non-private data can be used in the server to mitigate the degradation derived from data heterogeneity. We update the global model using this shared non-private data after every aggregation with only one epoch. Table 2 describes the result of this experiment. We update the entire network (F in Table 2) on the server. As $p$ increases, the initial accuracy increases, as expected. However, the personalized accuracy decreases under significant data heterogeneity ($s$=10). This result implies that boosting a single global model can hurt personalization, which may be considered more important than the initial performance. Therefore, we agree that the development of an excellent global model should consider the ability to be adequately fine-tuned or personalized.

To investigate the cause of personalization performance degradation, we hypothesize that unnecessary and disturbing information for personalization (i.e., the information of similar classes that a certain client does not have) is injected into the head, when a global model is trained on the server.[5] To capture this, we only update the body (B in Table 2) on the server by zeroing the learning rate corresponding to the head. By narrowing the update parts, the personalization degradation problem can be remedied significantly without affecting the initial accuracy. From this observation, we argue that training the head using shared data negatively impacts the personalization.[6]

---

[4]Non-private data are sampled randomly in our experiment, which can violate the FL environments. However, this experiment is conducted simply as a motivation for our study.

[5]We also investigate personalization of centralized trained models such as (Jiang et al., 2019), and the results are reported in Appendix C. Even in this case, training the head also has a negative impact on personalization.

[6]Zhuang et al. (2021) and Luo et al. (2021) posed a similar problem that the head can be easily biased because it is closest to the client's label distribution. Zhuang et al. (2021) proposed a divergence-aware predictor update module, and Luo et al. (2021) and Achituve et al. (2021) proposed a head calibration method.

# 5 FEDBABU: FEDERATED AVERAGING WITH BODY AGGREGATION AND BODY UPDATE

We propose a novel algorithm that learns a better single global model to be personalized efficiently. Inspired by prior studies on long-tailed recognition (Yu et al., 2020; Kang et al., 2019), fine-tuning (Devlin et al., 2019), and self-supervised learning (He et al., 2020), as well as our data-sharing experiment, we decouple the entire network into the body and the head. The body is trained for generalization and the head is then trained for specialization. We apply this idea to federated learning by *never* training the head in the federated training phase (i.e., developing a single global model) and by fine-tuning the head for personalization (in the evaluation process).

## 5.1 FROZEN HEAD IN THE CENTRALIZED SETTING

Before proposing our algorithm, we empirically demonstrate that a model with the initialized and fixed head (body in Figure 2) has comparable performance to a model that jointly learns the body and the head (full in Figure 2). Figure 2 depicts the test accuracy curve of MobileNet on CIFAR100 for various training scenarios in the centralized setting, where total epochs are 160 and learning rate starts with 0.1 and is decayed by a factor of 0.1 at 80 and 120 epochs. The blue line represents the accuracy when all layers are trained, the red line represents the accuracy when only the body of the model is trained, and the green line represents the accuracy when only the head of the model is trained. The fully trained model and the

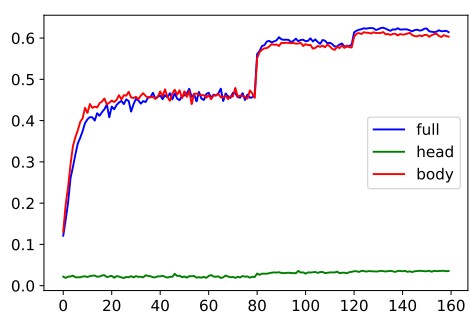

Figure 2: Test accuracy curves according to the update part in the centralized setting.

fixed head have almost the same performance, whereas the fixed body performs poorly. Thus, we claim that *the randomly initialized fixed head is acceptable, whereas the randomly initialized fixed body is unacceptable; initialized head is thought to serve as guidelines.* This characteristic is derived from the orthogonality on the head, as is explained in Appendix B. These results are the reasons we can update only the body during local training in FL settings.

## 5.2 FEDBABU ALGORITHM

Based on the insights and results in Section 5.1, we propose a new FL algorithm called FedBABU (**Fed**erated Averaging with **B**ody **A**ggregation and **B**ody **U**pdate). Only the body is trained, whereas the head is never trained during federated training; therefore, there is no need to aggregate the head. Formally, the model parameters $\theta$ are decoupled into the body (extractor) parameters $\theta_{ext}$ and the head (classifier) parameters $\theta_{cls}$. Note that $\theta_{cls}$ on any client is fixed with the head parameters of a randomly initialized global parameters $\theta_G^0$ until a single global model converges. This is implemented by zeroing the learning rate that corresponds to the head. *Intuitively, it is believed that the same fixed head on all clients serves as the same criteria on learning representations across all clients despite the passage of training time.*

**Algorithm 1** Training procedure of FedBABU.

1: initialize $\theta_G^0 = \{\theta_{G,ext}^0, \theta_{G,cls}^0\}$
2: **for** each round $k = 1, \cdots, $ K **do**
3:     $m \leftarrow max(\lfloor Nf \rfloor, 1)$
4:     $C^k \leftarrow$ random subset of $m$ clients
5:     **for** each client $C_i^k \in C^k$ in parallel **do**
6:         $\theta_i^k(0) \leftarrow \theta_G^{k-1} = \{\theta_{G,ext}^{k-1}, \theta_{G,cls}^0\}$
7:         $\theta_{i,ext}^k(\tau I_i^k) \leftarrow$ **ClientBodyUpdate**$(\theta_i^k(0), \tau)$
8:     **end for**
9:     $\theta_{G,ext}^k \leftarrow \sum_{i=1}^m \frac{n_{C_i^k}}{n} \theta_{i,ext}^k(\tau I_i^k), n = \sum_{i=1}^m n_{C_i^k}$
10: **end for**
11: **return** $\theta_G^K = \{\theta_{G,ext}^K, \theta_{G,cls}^0\}$
12: **function** CLIENTBODYUPDATE$(\theta_i^k, \tau)$
13:     $I_i^k \leftarrow \lceil \frac{n_{C_i^k}}{B} \rceil$
14:     **for** each local epoch $1, \cdots, \tau$ **do**
15:         **for** each iteration $1, \cdots, I_i^k$ **do**
16:             $\theta_{i,ext}^k \leftarrow SGD(\theta_{i,ext}^k, \theta_{G,cls}^0)$
17:         **end for**
18:     **end for**
19:     **return** $\theta_{i,ext}^k$
20: **end function**

Algorithm 1 describes the training procedure of FedBABU. All notations are explained in the FL training procedure paragraph in Section 3. Lines 3-4, Line 6, Line 7, and Line 9 correspond to the client sampling, broadcasting, local update, and aggregation stage, respectively. In the ClientBodyUpdate function, the local body parameter $\theta_{i,ext}^k$ is always updated based on the same head $\theta_{G,cls}^0$ (Line 16). Then, only the global body parameter $\theta_{G,ext}^k$ is aggregated (Line 9). Therefore, the final global parameter $\theta_G^K$ (Line 11) and the initialized global parameter $\theta_G^0$ (Line 1) have the same head parameter $\theta_{G,cls}^0$.

In this section, we first investigate the representation power after federated training (Section 5.2.1). Next, we demonstrate the ability related to personalization (Section 5.2.2 to Section 5.2.4). We further show our algorithm's applicability (Section 5.2.5).

### 5.2.1 REPRESENTATION POWER OF GLOBAL MODELS TRAINED BY FEDAVG AND FEDBABU

Higher initial accuracy is often required because some clients may not have data for personalization in practice. In addition, the representation power of the initial models is related to the performance of downstream or personal tasks. To capture the representation power, the "without (w/o) head" accuracy is calculated by replacing the trained head with the nearest template following previous studies (Kang et al., 2019; Snell et al., 2017; Raghu et al., 2019; Oh et al., 2021): (1) Using the trained global model and training data set on each client, the representations of each class are averaged into the template of each class. Then, (2) the cosine similarity between the test samples and the templates are measured and the test samples are classified into the nearest template's class. Templates are created for each client because raw data cannot be moved.

Table 3 describes the initial accuracy of FedAvg and FedBABU according to the existence of the head. Because all clients share the same criteria

Table 3: Initial accuracy of FedAvg and FedBABU on CIFAR100 under various settings with 100 clients. The trained head is replaced with the nearest template to calculate "w/o head" accuracy and MobileNet is used.

| FL settings | | | FedAvg | | FedBABU | |
|---|---|---|---|---|---|---|
| $s$ | $f$ | $\tau$ | w/ head | w/o head | w/ head | w/o head |
| 100 | 1.0 | 1 | 46.93±5.47 | 46.23±4.53 | 48.61±4.75 | 49.97±4.69 |
| | | 4 | 37.44±4.98 | 33.48±5.09 | 37.32±4.39 | 37.20±4.35 |
| | | 10 | 29.58±4.87 | 25.11±4.60 | 26.69±4.50 | 27.70±4.51 |
| | 0.1 | 1 | 39.07±5.22 | 36.69±5.82 | 41.02±4.99 | 41.19±4.96 |
| | | 4 | 35.39±4.58 | 32.58±4.37 | 36.77±4.47 | 36.61±4.64 |
| | | 10 | 28.18±4.83 | 24.34±4.58 | 29.38±4.74 | 29.36±4.46 |
| 50 | 1.0 | 1 | 45.68±5.50 | 53.87±5.39 | 47.19±4.77 | 55.70±5.48 |
| | | 4 | 36.05±4.04 | 42.65±4.76 | 37.27±5.25 | 45.25±5.45 |
| | | 10 | 29.57±4.29 | 34.13±4.44 | 28.43±4.72 | 36.19±4.93 |
| | 0.1 | 1 | 38.20±5.73 | 44.57±5.34 | 41.33±5.10 | 49.18±5.73 |
| | | 4 | 33.49±4.72 | 40.01±5.49 | 34.68±4.58 | 42.43±5.11 |
| | | 10 | 27.34±4.96 | 33.10±5.08 | 27.91±5.27 | 36.49±5.37 |
| 10 | 1.0 | 1 | 37.27±6.97 | 67.18±7.27 | 45.32±8.52 | 71.23±6.71 |
| | | 4 | 24.17±5.50 | 58.70±6.74 | 32.91±7.07 | 64.41±7.44 |
| | | 10 | 17.85±7.38 | 51.72±7.65 | 22.15±5.72 | 55.63±7.24 |
| | 0.1 | 1 | 29.12±7.11 | 60.42±7.89 | 35.05±7.63 | 65.98±6.43 |
| | | 4 | 21.14±6.86 | 54.91±6.72 | 25.67±7.31 | 59.44±6.43 |
| | | 10 | 14.40±5.64 | 50.25±6.27 | 18.50±7.82 | 54.93±7.85 |

and learn representations based on that criteria during federated training in FedBABU, improved representation power is expected.[7] When evaluated without the head, FedBABU has improved performance compared to FedAvg in all cases, particularly under large data heterogeneity. Specifically, when $s = 100$ (i.e., each client has most of the classes of CIFAR100), the performance gap depends on whether the head exists in FedAvg, but not in FedBABU. This means that the features through FedBABU are well-represented to the extent that there is nothing to be supplemented through the head. When $s$ is small, the performance improves when there is no head. Because templates are constructed based on the training data set for each client, it is not possible to classify test samples into classes that the client does not have. In other words, restrictions on the output distribution of each client lead to tremendous performance gains, even without training the global model. Therefore, it is expected that fine-tuning the global models with a strong representation can boost personalization.

### 5.2.2 PERSONALIZATION OF FEDAVG AND FEDBABU

Table 4: Personalized accuracy of FedAvg and FedBABU on CIFAR100 according to the fine-tuned part. The fine-tuning epochs is 5 and $f$ is 0.1.

(a) FedBABU.

| FL settings | | Update part for personalization | | |
|---|---|---|---|---|
| $s$ | $\tau$ | Body | Head | Full |
| 100 | 1 | 44.26±5.12 | 49.76±5.03 | 49.67±4.92 |
| | 4 | 39.61±4.68 | 44.74±5.02 | 44.74±5.10 |
| | 10 | 32.45±5.42 | 36.48±5.04 | 35.94±5.06 |
| 50 | 1 | 48.54±5.23 | 56.76±5.68 | 56.69±5.16 |
| | 4 | 41.27±5.04 | 49.45±5.41 | 49.55±5.58 |
| | 10 | 35.42±5.60 | 42.55±5.70 | 42.63±5.59 |
| 10 | 1 | 72.81±7.32 | 75.97±6.29 | 76.02±6.29 |
| | 4 | 69.12±6.70 | 70.74±6.47 | 71.00±6.63 |
| | 10 | 64.77±7.14 | 66.28±6.77 | 66.32±7.02 |

(b) FedAvg.

| FL settings | | Update part for personalization | | |
|---|---|---|---|---|
| $s$ | $\tau$ | Body | Head | Full |
| 100 | 1 | 41.00±5.35 | 43.18±5.34 | 43.92±5.55 |
| | 4 | 37.43±4.98 | 38.29±4.96 | 39.67±5.21 |
| | 10 | 30.62±4.95 | 31.92±5.04 | 33.13±5.22 |
| 50 | 1 | 43.61±5.54 | 47.51±5.61 | 49.55±5.36 |
| | 4 | 37.99±4.68 | 41.48±4.61 | 43.63±4.77 |
| | 10 | 32.20±4.92 | 36.06±5.31 | 38.09±5.17 |
| 10 | 1 | 56.00±8.66 | 69.70±7.88 | 71.24±7.82 |
| | 4 | 34.49±7.92 | 65.32±6.81 | 67.14±6.72 |
| | 10 | 27.94±6.96 | 60.24±6.16 | 62.67±6.52 |

---

[7]Appendix O provides the results of FedAvg with different learning rates between the head and the body, which shows the freezing the head completely is critical. Appendix P further provides class-wise analysis during federated training, which shows that freezing the head makes less damage to out-of-class during local updates.

To investigate the dominant factor for personalization of FedBABU, we compare the performance according to the fine-tuned part. Table 4a describes the results of this experiment. Global models are fine-tuned with five epochs based on the training data set of each client. It is shown that fine-tuning including the head (i.e., Head or Full in Table 4a) is better for personalization than body-partially fine-tuning (i.e., Body in Table 4a). It is noted that the computational costs can be reduced by fine-tuning only the head in the case of FedBABU without performance degradation.

However, in the case of FedAvg, a performance gap appears. Table 4b describes the personalized accuracy of FedAvg according to the fine-tuned parts. It is shown that fine-tuning the entire network (Full in Table 4b) is better for personalization than partial fine-tuning (Body or Head in Table 4b). Therefore, for FedAvg, it is recommended to personalize global models by fine-tuning the entire network for better performance. For consistency of the evaluation, for both FedAvg and FedBABU, we fine-tuned the entire model in this paper.

### 5.2.3 PERSONALIZATION PERFORMANCE COMPARISON

Table 5: Personalized accuracy comparison on CIFAR100 under various settings with 100 clients and MobileNet is used.

| FL settings | | | Personalized accuracy | | | | | | | |
|---|---|---|---|---|---|---|---|---|---|---|
| $s$ | $f$ | $\tau$ | FedBABU (Ours) | FedAvg (2017) | FedPer (2019) | LG-FedAvg (2020) | FedRep (2021) | Per-FedAvg (2020) | Ditto (2021) | Local-only |
| 100 | 1.0 | 1 | **55.79**±4.57 | 51.93±5.19 | 51.95±5.30 | 53.01±5.26 | 18.29±3.59 | 47.09±7.35 | 39.36±4.78 | 20.60±3.15 |
| | | 4 | **44.49**±4.91 | 42.66±5.09 | 40.87±5.05 | 43.09±4.74 | 15.32±3.79 | 39.07±7.59 | 31.58±5.00 | |
| | | 10 | 33.20±4.54 | 34.62±4.97 | 32.91±4.97 | **34.64**±5.03 | 13.45±3.26 | 30.22±6.59 | 21.18±4.54 | |
| | 0.1 | 1 | **49.67**±4.92 | 43.92±5.55 | 45.17±4.70 | 40.91±5.50 | 23.84±3.92 | 48.10±7.42 | 45.46±4.73 | |
| | | 4 | **44.74**±5.10 | 39.67±5.21 | 39.30±4.92 | 37.87±4.99 | 16.01±3.48 | 33.70±7.04 | 32.46±5.42 | |
| | | 10 | **35.94**±5.06 | 33.13±5.22 | 32.08±4.97 | 30.08±5.34 | 11.11±3.13 | 25.82±5.83 | 23.96±4.64 | |
| 50 | 1.0 | 1 | **61.09**±4.91 | 57.84±5.08 | 57.16±5.26 | 58.44±5.53 | 24.75±5.02 | 43.75±7.94 | 42.70±5.46 | 28.02±4.01 |
| | | 4 | **51.56**±5.04 | 47.17±4.26 | 48.89±5.40 | 47.78±4.72 | 21.55±4.36 | 37.59±7.87 | 36.57±5.11 | |
| | | 10 | **42.09**±5.12 | 40.59±5.23 | 39.90±5.54 | 40.32±4.70 | 19.92±4.50 | 28.75±6.40 | 27.27±5.04 | |
| | 0.1 | 1 | **56.69**±5.16 | 49.55±5.36 | 51.63±5.27 | 42.64±5.55 | 32.88±5.09 | 43.96±7.40 | 43.22±5.82 | |
| | | 4 | **49.55**±5.58 | 43.63±4.77 | 46.31±5.63 | 38.54±4.71 | 21.13±3.96 | 28.67±6.98 | 31.65±5.08 | |
| | | 10 | **42.63**±5.59 | 38.09±5.17 | 39.81±4.88 | 30.79±6.12 | 15.15±4.01 | 21.64±6.16 | 22.16±4.67 | |
| 10 | 1.0 | 1 | **79.17**±6.51 | 77.46±5.78 | 74.71±6.35 | 77.49±5.60 | 61.28±8.27 | 36.59±8.98 | 65.33±7.49 | 61.52±7.22 |
| | | 4 | **74.60**±6.69 | 70.41±6.83 | 65.61±7.13 | 69.97±6.42 | 50.59±7.94 | 18.31±10.57 | 64.47±7.45 | |
| | | 10 | **66.64**±6.84 | 63.51±7.38 | 59.71±7.35 | 61.50±7.28 | 42.13±7.53 | 11.54±8.87 | 51.68±7.44 | |
| | 0.1 | 1 | **76.02**±6.29 | 71.24±7.82 | 69.36±6.77 | 51.75±9.32 | 60.13±7.72 | 31.21±11.66 | 31.91±15.10 | |
| | | 4 | **71.00**±6.63 | 67.14±6.72 | 62.62±7.63 | 35.80±10.55 | 45.91±7.68 | 14.34±9.51 | 23.70±15.84 | |
| | | 10 | **66.32**±7.02 | 62.67±6.52 | 59.50±7.33 | 25.04±12.02 | 34.30±7.84 | 9.17±6.95 | 14.24±15.67 | |

We compare FedBABU with existing methods from the perspective of personalization. Appendix A presents details of the evaluation procedure and implementations of each algorithm. Table 5 describes the personalized accuracy of various algorithms. Notably, FedAvg is a remarkably strong baseline, overwhelming other recent personalized FL algorithms on CIFAR100 in most cases.[8] These results are similar to recent trends in the field of meta-learning, where fine-tuning based on well-trained representations overwhelms advanced few-shot classification algorithms for heterogeneous tasks (Chen et al., 2019a; 2020a; Dhillon et al., 2019; Tian et al., 2020). FedBABU (ours) further outshines FedAvg; it is believed that enhancing representations by freezing the head improves the performance.

In detail, when $\tau=1$, the performance gap between FedBABU and FedRep demonstrates the importance of fixing the head across clients. Note that when $\tau=1$, if there is no training process on the head in FedRep, FedRep is reduced to FedBABU. Moreover, we attribute the performance degradation of FedRep to the low epochs of training on the body. In addition, the efficiency of personalized FL algorithms increases when all clients participate; therefore, FedPer (Arivazhagan et al., 2019) assumes the activation of all clients during every communication round. The Per-FedAvg algorithm might suffer from dividing the test set into a support set and query set when there are many classes and from small fine-tuning iterations. In summary, FedAvg are comparable and better than even recent personalized FL algorithms such as Per-FedAvg and Ditto, and FedBABU (ours) achieves the higher performance than FedAvg and other decoupling algorithms.

---

[8]Similar results were reported in the recent paper (Collins et al., 2021), showing that FedAvg beats recent personalized algorithms (Smith et al., 2017; Fallah et al., 2020; Liang et al., 2020; Hanzely & Richtárik, 2020; Deng et al., 2020; Li et al., 2021; Arivazhagan et al., 2019) in many cases. Appendix N further discusses the effectiveness of FedAvg.

### 5.2.4 PERSONALIZATION SPEED OF FEDAVG AND FEDBABU

Table 6: Performance according to the fine-tune epochs (FL setting: $f$=0.1, and $\tau$=10).

| $s$ | Algorithm | Fine-tune epochs ($\tau_f$) | | | | | | | | |
|-----|-----------|------------|---|---|---|---|---|---|---|---|
| | | 0 (Initial) | 1 | 2 | 3 | 4 | 5 | 10 | 15 | 20 |
| 50 | FedAvg | 27.34±4.96 | 29.17±5.01 | 32.39±4.77 | 34.97±5.13 | 36.78±5.13 | 38.09±5.17 | 40.56±5.43 | 41.20±5.51 | 40.86±5.13 |
| | FedBABU | 27.91±5.27 | 35.20±5.58 | 40.60±5.47 | 42.12±5.61 | 42.74±5.60 | 42.63±5.59 | 41.94±5.68 | 41.19±5.52 | 40.61±5.28 |
| 10 | FedAvg | 14.40±5.64 | 27.43±6.46 | 48.63±7.30 | 58.08±6.11 | 61.27±6.15 | 62.67±6.52 | 63.91±6.49 | 64.56±6.45 | 64.89±6.53 |
| | FedBABU | 18.50±7.82 | 63.29±7.55 | 66.05±6.93 | 66.10±6.54 | 66.40±7.24 | 66.32±7.02 | 66.07±7.57 | 66.24±7.67 | 66.32±7.71 |

We investigate the personalization speed of FedAvg and FedBABU by controlling the fine-tuning epochs $\tau_f$ in the evaluation process. Table 6 describes the initial (when $\tau_f$ is 0) and personalized (otherwise) accuracy of FedAvg and FedBABU. Here, one epoch is equal to 10 updates in our case because each client has 500 training samples and the batch size is 50. It has been shown that FedAvg requires sufficient epochs for fine-tuning, as reported in Jiang et al. (2019). Notably, FedBABU achieves better accuracy with a small number of epochs, which means that FedBABU can personalize accurately and rapidly, especially when fine-tuning is either costly or restricted. This characteristic is explained further based on cosine similarity in Appendix K.

### 5.2.5 BODY AGGREGATION AND BODY UPDATE ON THE FEDPROX

Table 7: Initial and personalized accuracy of FedProx and FedProx+BABU with $\mu$=0.01 on CIFAR100 with 100 clients and $f$=0.1.

| Algorithm | $\tau$ | $s$=100 (heterogeneity ↓) | | $s$=50 | | $s$=10 (heterogeneity ↑) | |
|-----------|--------|-----------|--------------|-----------|--------------|-----------|--------------|
| | | Initial | Personalized | Initial | Personalized | Initial | Personalized |
| FedProx | 1 | 46.52±4.56 | 50.95±4.65 | 42.20±4.90 | 51.29±5.20 | 28.16±9.00 | 66.39±7.79 |
| | 4 | 36.54±4.74 | 39.83±4.71 | 33.59±4.80 | 40.17±5.11 | 18.20±7.62 | 41.56±9.34 |
| | 10 | 28.63±4.40 | 31.90±4.16 | 26.88±4.59 | 32.92±5.00 | 13.62±7.73 | 43.48±9.32 |
| FedProx +BABU | 1 | 48.53±5.15 | 57.44±4.72 | 46.25±5.31 | 63.12±5.25 | 33.13±8.11 | 78.86±5.70 |
| | 4 | 37.17±4.41 | 45.26±4.76 | 33.86±5.44 | 50.18±5.14 | 22.94±9.90 | 75.71±5.33 |
| | 10 | 27.79±3.95 | 35.68±4.34 | 27.48±5.22 | 42.37±6.10 | 15.66±8.29 | 67.15±7.10 |

FedProx (Li et al., 2018) regularizes the distance between a global model and local models to prevent local models from deviating. The degree of regularization is controlled by $\mu$. Note that when $\mu$ is 0.0, FedProx is reduced to FedAvg. Table 7 describes the initial and personalized accuracy of FedProx and FedProx+BABU with $\mu$=0.01 when $f$=0.1, and Appendix M reports the results when $f$=1.0. The global models trained by FedProx reduce the personalization capabilities compared to FedAvg (refer to Table 1 when $f$=0.1), particularly under realistic FL settings. We adapt the body aggregation and body update idea to the FedProx algorithm, referred to as FedProx+BABU, which performs better than the personalization of FedAvg. Furthermore, FedProx+BABU improves personalized performance compared to FedBABU (refer to Table 5 when $f$=0.1). This means that the regularization of the body is still meaningful. Our algorithm and various experiments suggest future directions of federated learning: *Which parts should be federated and enhanced? Representation!*

## 6 CONCLUSION

In this study, we focused on how to train a good federated global model for the purpose of personalization. Namely, this problem boils down to how to pre-train a better backbone in a federated learning manner for downstream personal tasks. We first showed that existing methods to improve a global model (e.g., data sharing or regularization) could reduce ability to personalize. To mitigate this personalization performance degradation problem, we decoupled the entire network into the body, related to generality, and the head, related to personalization. Then, we demonstrated that training the head creates this problem. Furthermore, we proposed the FedBABU algorithm, which learns a shared global model that can rapidly adapt to heterogeneous data on each client. FedBABU only updates the body in the local update stage and only aggregates the body in the aggregation stage, thus developing a single global model with a strong representation. This global model can be efficiently personalized by fine-tuning each client's model using its own data set. Extensive experimental results showed that FedBABU overwhelmed various personalized FL algorithms. Our improvement emphasizes the importance of federating and enhancing the representation for FL.

ACKNOWLEDGMENTS

This work was supported by Institute of Information & communications Technology Planning & Evaluation (IITP) grant funded by the Korea government (MSIT) [No.2019-0-00075, Artificial Intelligence Graduate School Program (KAIST)] and [No. 2021-0-00907, Development of Adaptive and Lightweight Edge-Collaborative Analysis Technology for Enabling Proactively Immediate Response and Rapid Learning].

REPRODUCIBILITY STATEMENT

Our code is available at `https://github.com/jhoon-oh/FedBABU`. For convenience reproducibility, shell files of each algorithm are also included.

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

# A    IMPLEMENTATION DETAIL

Our implementations are based on the code presented by (Liang et al., 2020).[9] All the reported results are based on the last model. For computation costs, we used a single TITAN RTX and the entire training time of FedBABU on CIFAR100 using MobileNet takes ~2 hours when $f$=1.0 and $\tau$=1. Therefore, as $f$ decreases and $\tau$ increases, it takes less time.

## A.1    ARCHITECTURES

In our experiments, we employ the 3convNet for EMNIST, 4convNet for CIFAR10, MobileNet for CIFAR100, and ResNet for CIFAR100. 3convNet and 4convNet are described below. MobileNet (Howard et al., 2017) includes depthwise convolution and pointwise convolution, and ResNet (He et al., 2016) includes skip connection.[10] Please refer to each paper in detail.

```
3convNet(
  (features): Sequential(
    (0): Sequential(
      (0): Conv2d(1, 64, kernel_size=(3, 3), stride=(1, 1), padding=(1, 1))
      (1): BatchNorm2d(64, eps=1e-05, momentum=0.1, affine=True, track_running_stats=False)
      (2): ReLU()
      (3): MaxPool2d(kernel_size=2, stride=2, padding=0, dilation=1, ceil_mode=False)
    )
    (1): Sequential(
      (0): Conv2d(64, 64, kernel_size=(3, 3), stride=(1, 1), padding=(1, 1))
      (1): BatchNorm2d(64, eps=1e-05, momentum=0.1, affine=True, track_running_stats=False)
      (2): ReLU()
      (3): MaxPool2d(kernel_size=2, stride=2, padding=0, dilation=1, ceil_mode=False)
    )
    (2): Sequential(
      (0): Conv2d(64, 64, kernel_size=(3, 3), stride=(1, 1), padding=(1, 1))
      (1): BatchNorm2d(64, eps=1e-05, momentum=0.1, affine=True, track_running_stats=False)
      (2): ReLU()
      (3): MaxPool2d(kernel_size=2, stride=2, padding=0, dilation=1, ceil_mode=False)
    )
  )
  (linear): Linear(in_features=576, out_features=62, bias=True)
)

4convNet(
  (features): Sequential(
    (0): Sequential(
      (0): Conv2d(3, 64, kernel_size=(3, 3), stride=(1, 1), padding=(1, 1))
      (1): BatchNorm2d(64, eps=1e-05, momentum=0.1, affine=True, track_running_stats=False)
      (2): ReLU()
      (3): MaxPool2d(kernel_size=2, stride=2, padding=0, dilation=1, ceil_mode=False)
    )
    (1): Sequential(
      (0): Conv2d(64, 64, kernel_size=(3, 3), stride=(1, 1), padding=(1, 1))
      (1): BatchNorm2d(64, eps=1e-05, momentum=0.1, affine=True, track_running_stats=False)
      (2): ReLU()
      (3): MaxPool2d(kernel_size=2, stride=2, padding=0, dilation=1, ceil_mode=False)
    )
    (2): Sequential(
      (0): Conv2d(64, 64, kernel_size=(3, 3), stride=(1, 1), padding=(1, 1))
      (1): BatchNorm2d(64, eps=1e-05, momentum=0.1, affine=True, track_running_stats=False)
      (2): ReLU()
      (3): MaxPool2d(kernel_size=2, stride=2, padding=0, dilation=1, ceil_mode=False)
    )
    (3): Sequential(
      (0): Conv2d(64, 64, kernel_size=(3, 3), stride=(1, 1), padding=(1, 1))
      (1): BatchNorm2d(64, eps=1e-05, momentum=0.1, affine=True, track_running_stats=False)
      (2): ReLU()
      (3): MaxPool2d(kernel_size=2, stride=2, padding=0, dilation=1, ceil_mode=False)
    )
  )
  (linear): Linear(in_features=256, out_features=10, bias=True)
)
```

---

[9]https://github.com/pliang279/LG-FedAvg

[10]We use two architectures from https://github.com/kuangliu/pytorch-cifar

## A.2 Datasets

We used two public datasets, the CIFAR (Krizhevsky et al., 2009)[11] and EMNIST (Cohen et al., 2017)[12], for performance evaluation. The composition of each dataset is summarized in Table 8. We applied linear transformations such as horizontal flipping and random cropping.

Table 8: Composition of CIFAR and EMNIST.

| Dataset | # of Training | # of Validation | # of Classes |
|---|---|---|---|
| CIFAR-10 | 50,000 | 10,000 | 10 |
| CIFAR-100 | 50,000 | 10,000 | 100 |
| EMNIST | 731,668 | 82,587 | 62 |

## A.3 Hyperparameters

For **LG-FedAvg**, FedAvg models corresponding to the FL settings in Table 3 are used as an initialization. It is then trained about a quarter of FedAvg training with a learning rate of 0.001. For **FedRep**, the number of local epochs for head and body updates were set to $\tau$ and 1, respectively. For **Per-FedAvg**, we used the first-order (FO) version, and the $\beta$ value was set as the learning rate value corresponding to the communication round. For **Ditto**, the $\lambda$ used to control the regularization term was set to 0.75.

## A.4 Evaluation of FL algorithms

In FL algorithms, because the parts that need to be shared over all clients after federated training may not be shared due to client fraction, we evaluate these algorithms as follows:

---

**Algorithm 2** Evaluation procedure of FedAvg, FedBABU, FedPer, LG-FedAvg, and FedRep.

---
1:  initlist = [ ]                                                                       (If $alg$ is FedAvg or FedBABU)
2:  perlist = [ ]
3:  **for** each client $i \in [1, N]$ in parallel **do**
4:      $\theta_{i,s}(0) \leftarrow \theta_{G,s}^{K}$
5:      $Acc_{init} \leftarrow$ **Eval**$(\theta_i(0); D_i^{ts})$                          (If $alg$ is FedAvg or FedBABU)
6:      initlist.append($Acc_{init}$)                                                     (If $alg$ is FedAvg or FedBABU)
7:      $\theta_i(\tau_f I_i) \leftarrow$ **Fine-tune**$(alg, \theta_i(0), \tau_f; D_i^{tr})$
8:      $Acc_{per} \leftarrow$ **Eval**$(\theta_i(\tau_f I_i); D_i^{ts})$
9:      perlist.append($Acc_{per}$)
10: **end for**
11: **return** initlist, perlist

12: **function** Fine-tune$(alg, \theta_i, \tau_f)$
13:     $I_i \leftarrow \lceil \frac{n_{C_i}}{B} \rceil$
14:     **for** each fine-tune epoch $1, \cdots, \tau_f$ **do**
15:         **for** each iteration $1, \cdots, I_i$ **do**
16:             $\theta_i \leftarrow SGD(\theta_i)$                                       (If $alg$ is FedAvg or FedBABU or FedPer or LG-FedAvg)
17:             $\theta_{i,cls} \leftarrow SGD(\theta_i)$                                 (If $alg$ is FedRep)
18:         **end for**
19:     **end for**
20:     **for** each iteration $1, \cdots, I_i$ **do**                                    (If $alg$ is FedRep)
21:         $\theta_{i,ext} \leftarrow SGD(\theta_i)$
22:     **end for**
23:     **return** $\theta_i$
24: **end function**

---

where $\theta_{G,s}^{K}$ indicates the shared parts after federated training (i.e., full in the cases of FedAvg and FedBABU, body in the cases of FedPer and FedRep, and head in the case of LG-FedAvg). All notations are explained in the evaluation paragraph in Section 3. Because communication round $k$ is not related to evaluation, it is emitted. For FedAvg and FedBABU, *initial accuracy* is also calculated. Although all clients have different personalized models because some parts are not shared after broadcasting except for FedAvg and FedBABU, it is not completely personalized because there is a common part $\theta_{G,s}^{K}$. Therefore, we fine-tune all algorithms. FedRep trains models sequentially (Line 17 and 21), while others jointly (Line 16). Finally, the mean and standard deviation of initlist and perlist are reported as *initial accuracy* and *personalized accuracy* in our paper.

---

[11]https://www.cs.toronto.edu/ kriz/cifar.html
[12]https://www.nist.gov/itl/products-and-services/emnist-dataset

## B ORTHOGONALITY OF RANDOM INITIALIZATION

We hypothesize that the orthogonal initialization (Saxe et al., 2013) on the head, through which the row vectors of the classifier parameters $\theta_{cls}$ are orthogonal, is the best for the body-only update rule. At each update, the output of the extractor is pulled toward its label's row of $\theta_{cls}$ and pushed apart from the other rows of $\theta_{cls}$ through backpropagation. Therefore, orthogonal weight initialization can separate the outputs of the extractor well based on their class labels. Note that distribution-based random initializations, such as Xaiver (Glorot & Bengio, 2010) and He (He et al., 2015), have almost orthogonal rows because the orthogonality of two random vectors is observed in high dimensions with high probability (Lezama et al., 2018).

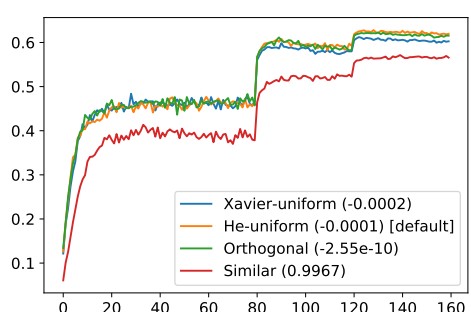

Figure 3: Test accuracy curves according to the initialization method when the body is only trained in the centralized setting. The values in parentheses indicate the average of cosine similarities between row vectors.

Figure 3 shows test accuracy curves for many different initialization methods when only the body is trained in the centralized setting.[13] When the row vectors are initialized similarly, a convergence gap appears (green and red lines in Figure 3), as expected. More experiments in the centralized setting are reported in subsection F.1 and Appendix G. Based on these empirical and theoretical results, we used He (uniform) initialization, the default setting in PyTorch (Paszke et al., 2019), in our paper.

This characteristic can be proved. Xavier (Glorot & Bengio, 2010) and He (He et al., 2015) are representative distribution-based random initialization methods, controlling the variance of parameters using network size to avoid the exploding/vanishing gradient problem. Two initialization methods use uniform and normal distribution. In detail, each element follows $\mathcal{U}(-a, a)$ or $\mathcal{N}(0, \sigma^2)$ independently, where $a$ and $\sigma$ are defined by non-linearity (such as ReLU and Tanh) and layer input-output size. Because we used the ReLU non-linear function in our implementations, we calculate $a$ and $\sigma$ by multiplying $\sqrt{2}$, for scaling of ReLU non-linearity.

Because we focus on initialization on the head, we specify $\theta_{cls} \in \mathbb{R}^{C \times d}$, where $C$ is the number of classes (i.e., layer output size) and $d$ is the dimension of the last representation (i.e., layer input size). Namely, each element random variable $\{w_{i,j}\}_{i \in [1,C], j \in [1,d]}$ follows $\mathcal{U}(-a, a)$ or $\mathcal{N}(0, \sigma^2)$ independently.

When $w_{i,j} \sim \mathcal{U}(-a, a)$, in Xavier initialization, $a$ is defined as $\sqrt{2} \times \sqrt{\frac{6}{d+C}}$. Similarly, in He initialization, $a$ is defined as $\sqrt{2} \times \sqrt{\frac{3}{d}}$ or $\sqrt{2} \times \sqrt{\frac{3}{C}}$ depending on which to be preserved, forward pass or backward pass. When $w_{i,j} \sim \mathcal{N}(0, \sigma^2)$, in Xavier initialization, $\sigma$ is defined as $\sqrt{2} \times \sqrt{\frac{2}{d+C}}$. Similarly, in He initialization, $\sigma$ is defined as $\sqrt{2} \times \sqrt{\frac{1}{d}}$ or $\sqrt{2} \times \sqrt{\frac{1}{C}}$ depending on which to be preserved, forward pass or backward pass. However, It is noted that $a$ in the uniform distribution and $\sigma$ in the normal distribution do not affect proof on orthogonality, therefore we keep the form of $\mathcal{U}(-a, a)$ and $\mathcal{N}(0, \sigma^2)$ for simplicity.

We first consider $2d$ independent random variables $\{w_{i,j}\}_{i \in \{p,q\}, j \in [1,d]}$ for any $p \neq q \in [1, C]$. Then, we can define $d$ independent random variables $\{w_{p,j} w_{q,j}\}_{j \in [1,d]}$ from the above $2d$ independent random variables. Then, for all $j \in [1, d]$, $\mathbb{E}[w_{p,j} w_{q,j}] = \mathbb{E}[w_{p,j}] \mathbb{E}[w_{q,j}] = 0$ and $\mathbb{V}[w_{p,j} w_{q,j}] = \mathbb{V}[w_{p,j}] \mathbb{E}[w_{q,j}]^2 + \mathbb{V}[w_{p,j}] \mathbb{V}[w_{q,j}] + \mathbb{E}[w_{p,j}]^2 \mathbb{V}[w_{q,j}] = \mathbb{V}[w_{p,j}] \mathbb{V}[w_{q,j}]$ because each element in $\theta_{cls}$ follows $\mathcal{U}(-a, a)$ or $\mathcal{N}(0, \sigma^2)$ independently. Let $S_d = \frac{1}{d} \sum_{j=1}^{d} w_{p,j} w_{q,j}$. Then, by the Weak Law of Large Numbers, $\lim_{d \to \infty} P(|S_d - \mathbb{E}[S_d]| > \epsilon) = \lim_{d \to \infty} P(|S_d| > \epsilon) = 0$ for all $\epsilon > 0$ because $\{w_{p,j} w_{q,j}\}_{j \in [1,d]}$ are independent random variables and there is $V$ such that $\mathbb{V}[w_{p,j} w_{q,j}] < V$ for all $j$.

---

[13]All initialization methods except for 'similar' are provided by PyTorch (Paszke et al., 2019). Here, 'similar' initialization is implemented using a uniform distribution on [0.45, 0.55], and then each row vector is unitized by dividing the norm corresponding to the row vector.

## C    PERSONALIZATION OF THE CENTRALIZED MODEL

In (Jiang et al., 2019), they showed that localizing centralized initial models is harder. Similarly, we investigate the personalization of the centralized model under different heterogeneity. First, we train two models with 10 epochs, one is entirely updated (F in Table 9), and another is body-partially updated (B in Table 9) using all training data set. Then, the trained model is broadcast to each client and then evaluated. Table 9 describes the results of this experiment. It is demonstrated that if models are updated body-partially, then personalization ability does not hurt even under centralized training.

Table 9: Personalization of the centralized models. F is trained entirely, and B is trained body-partially in the centralized setting.

| $s$ | Model | Fine-tune epochs ($\tau_f$) | | | | | |
|---|---|---|---|---|---|---|---|
| | | 0 (Initial) | 1 | 2 | 3 | 4 | 5 |
| 100 | F | $40.25\pm4.75$ | $40.77\pm4.72$ | $41.80\pm4.86$ | $42.43\pm4.78$ | $43.32\pm4.72$ | $44.15\pm4.61$ |
| | B | $39.64\pm4.96$ | $43.37\pm5.41$ | $47.66\pm5.08$ | $50.16\pm5.24$ | $51.09\pm5.00$ | $52.15\pm5.04$ |
| 50 | F | $37.92\pm5.48$ | $38.93\pm5.52$ | $40.68\pm5.48$ | $42.40\pm5.33$ | $43.86\pm5.39$ | $45.27\pm5.43$ |
| | B | $37.18\pm5.21$ | $43.66\pm5.34$ | $51.14\pm5.49$ | $54.67\pm5.06$ | $56.34\pm4.82$ | $57.36\pm5.21$ |
| 10 | F | $25.46\pm6.23$ | $28.04\pm6.88$ | $33.43\pm7.32$ | $39.56\pm7.66$ | $44.74\pm7.43$ | $50.32\pm7.05$ |
| | B | $23.86\pm5.08$ | $48.63\pm5.39$ | $69.14\pm5.78$ | $74.32\pm5.35$ | $75.90\pm5.49$ | $76.47\pm5.48$ |

## D    EFFECT OF MOMENTUM DURING LOCAL UPDATES ON PERFORMANCE

We investigate the effect of momentum during local updates on the performance of FedAvg and FedBABU. Table 10 describes the initial and personalized accuracy according to the momentum during local updates. The momentum for personalization fine-tuning is the same as the momentum in federated training. In both cases of FedAvg and FedBABU, appropriate momentum improves the performance, especially personalized accuracy. However, when the extreme momentum ($m$=0.99) is used, FedAvg completely loses the ability to train models, while FedBABU has robust performance.

Table 10: Initial and personalized accuracy according to momentum ($m$) during local updates under a realistic FL setting ($N$=100, $f$=0.1, and $\tau$=10).

| $m$ | Initial accuracy | | | | | |
|---|---|---|---|---|---|---|
| | $s = 100$ | | $s = 50$ | | $s = 10$ | |
| | FedAvg | FedBABU | FedAvg | FedBABU | FedAvg | FedBABU |
| 0.0 | $26.07\pm3.83$ | $26.38\pm3.98$ | $24.71\pm4.15$ | $24.72\pm4.67$ | $15.44\pm7.16$ | $13.66\pm6.15$ |
| 0.5 | $26.63\pm4.61$ | $27.01\pm4.62$ | $26.27\pm4.87$ | $27.58\pm4.75$ | $16.60\pm6.14$ | $17.98\pm6.55$ |
| 0.9 | $28.18\pm4.83$ | $29.38\pm4.74$ | $27.34\pm4.96$ | $27.91\pm5.27$ | $14.40\pm5.64$ | $18.50\pm7.82$ |
| 0.99 | $19.84\pm4.12$ | $30.62\pm4.60$ | $1.00\pm1.40$ | $30.01\pm5.19$ | $1.00\pm3.32$ | $15.68\pm7.11$ |
| $m$ | Personalized accuracy | | | | | |
| | $s = 100$ | | $s = 50$ | | $s = 10$ | |
| | FedAvg | FedBABU | FedAvg | FedBABU | FedAvg | FedBABU |
| 0.0 | $28.46\pm4.05$ | $30.24\pm4.39$ | $30.02\pm4.65$ | $33.43\pm5.26$ | $40.80\pm9.00$ | $59.85\pm7.41$ |
| 0.5 | $30.44\pm4.37$ | $32.70\pm4.65$ | $33.64\pm4.78$ | $38.65\pm5.14$ | $54.32\pm8.33$ | $67.34\pm6.32$ |
| 0.9 | $33.13\pm5.22$ | $35.94\pm5.06$ | $38.09\pm5.17$ | $42.63\pm5.59$ | $62.67\pm6.52$ | $66.32\pm7.02$ |
| 0.99 | $23.88\pm4.34$ | $34.17\pm4.75$ | $1.12\pm1.48$ | $42.17\pm5.32$ | $1.00\pm3.32$ | $61.02\pm7.84$ |

## E    EFFECT OF MASSIVENESS ON PERFORMANCE

We increase the number of clients from 100 to 500, and then each client has 100 training data and 20 test data. Table 11 describes the initial and personalized accuracy of FedAvg and FedBABU on CIFAR100 under various FL settings with 500 clients. The performance of FedAvg is slightly better than that of FedBABU in many cases; however, FedBABU overwhelms FedBABU in the case of extreme FL setting ($s$=10, $f$=0.1, and $\tau$=10). More interestingly, if each client has a few data samples, then local epochs $\tau$ should be more than 1. It is thought that massiveness makes data size insufficient on each client (when the total number of data is fixed) and brings out other problems.

Table 11: Initial and personalized accuracy of FedAvg and FedBABU on CIFAR100 under various settings with 500 clients. The used network is MobileNet.

| FL settings | | | Initial accuracy | | Personalized accuracy | |
|---|---|---|---|---|---|---|
| $s$ | $f$ | $\tau$ | FedAvg | FedBABU | FedAvg | FedBABU |
| 100 | 1.0 | 1 | 13.71±7.58 | 11.85±7.13 | 16.12±7.99 | 15.00±7.54 |
| | | 4 | 18.98±8.57 | 16.79±8.38 | 21.17±9.05 | 19.94±8.64 |
| | | 10 | 15.70±8.18 | 15.66±8.23 | 17.54±8.91 | 18.47±8.61 |
| | 0.1 | 1 | 14.59±7.72 | 10.53±7.25 | 16.46±8.27 | 13.90±7.96 |
| | | 4 | 18.90±8.33 | 17.36±8.85 | 20.88±8.85 | 20.30±9.27 |
| | | 10 | 16.79±7.90 | 14.38±8.14 | 18.22±8.37 | 16.76±8.33 |
| 50 | 1.0 | 1 | 13.67±8.05 | 11.04±7.01 | 19.29±9.31 | 17.39±8.11 |
| | | 4 | 18.36±8.60 | 16.60±8.43 | 22.76±9.05 | 23.15±9.51 |
| | | 10 | 15.95±8.57 | 14.81±8.24 | 19.57±9.28 | 21.38±9.78 |
| | 0.1 | 1 | 14.58±8.33 | 10.72±6.81 | 19.65±9.14 | 16.05±8.11 |
| | | 4 | 18.28±8.79 | 16.79±9.04 | 22.39±9.78 | 23.57±9.79 |
| | | 10 | 15.79±8.21 | 15.07±7.97 | 18.52±8.69 | 20.83±8.90 |
| 10 | 1.0 | 1 | 9.91±7.14 | 7.41±6.55 | 37.62±11.76 | 35.02±10.71 |
| | | 4 | 15.34±8.49 | 12.67±7.58 | 39.91±12.43 | 49.07±11.13 |
| | | 10 | 12.99±7.98 | 12.93±8.15 | 35.47±11.29 | 48.07±12.25 |
| | 0.1 | 1 | 10.96±7.22 | 8.42±6.58 | 38.95±11.63 | 35.13±10.85 |
| | | 4 | 15.03±8.27 | 13.99±7.82 | 37.92±11.64 | 49.96±11.94 |
| | | 10 | 12.39±7.65 | 13.29±7.40 | 32.41±10.91 | 49.83±11.74 |

# F    RESULTS OF 4CONVNET ON CIFAR10

## F.1    ACCURACY CURVES IN THE CENTRALIZED SETTING

When 4convNet is used on CIFAR10, a similar tendency appears. Comparing Figure 2 and Figure 4, the importance of learning representation is emphasized more when tasks are difficult. In addition, Figure 5 shows the necessity of orthogonality when the body is only trained and approximation of random initialization to the orthogonal initialization using 4convNet on CIFAR10.

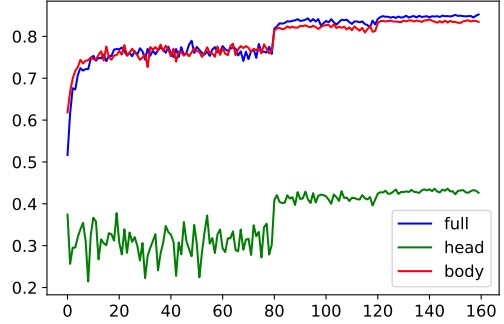

Figure 4: Test accuracy curves of 4convNet on CIFAR10 according to the update part in the centralized setting.

Figure 5: Test accuracy curves of 4convNet on CIFAR10 according to the initialization method when the body is only trained.

## F.2    EXPERIMENTAL RESULTS

We also evaluate algorithms on CIFAR10 using 4convNet. Table 12 and Table 13 are the experiment results corresponding to the Table 3 and Table 5 in the main paper, respectively. The tendency of performance comparison and the superiority of our algorithm appears similarly when using 4convNet on CIFAR10.

Table 12: Initial accuracy of FedAvg and FedBABU according to the existence of classifier on CIFAR10 under various settings with 100 clients. The used network is 4convNet.

| FL settings | | | FedAvg | | FedBABU | |
|---|---|---|---|---|---|---|
| $s$ | $f$ | $\tau$ | w/ classifier | w/o classifier | w/ classifier | w/o classifier |
| 10 | 1.0 | 1 | 68.89±6.35 | 74.41±6.25 | 71.50±7.35 | 75.00±7.46 |
| | | 4 | 61.56±6.84 | 66.17±7.44 | 68.40±6.67 | 71.77±7.29 |
| | | 10 | 60.06±5.68 | 65.21±7.94 | 64.66±6.00 | 69.04±6.68 |
| | 0.1 | 1 | 63.79±7.59 | 69.48±6.72 | 69.36±5.61 | 71.21±6.87 |
| | | 4 | 59.65±8.03 | 66.11±7.37 | 66.80±7.43 | 69.27±8.05 |
| | | 10 | 57.32±6.76 | 63.19±6.81 | 62.68±7.06 | 67.58±6.42 |
| 5 | 1.0 | 1 | 66.56±9.39 | 80.73±8.27 | 68.54±8.45 | 80.53±7.99 |
| | | 4 | 52.60±7.79 | 70.53±9.15 | 63.79±9.12 | 78.34±8.30 |
| | | 10 | 51.55±8.23 | 71.03±8.77 | 61.10±9.80 | 77.13±7.10 |
| | 0.1 | 1 | 55.13±8.57 | 71.27±10.03 | 63.44±9.82 | 77.64±8.26 |
| | | 4 | 48.77±8.54 | 68.64±11.36 | 59.93±8.32 | 74.61±8.81 |
| | | 10 | 46.32±10.52 | 67.07±8.77 | 55.55±9.90 | 73.29±9.25 |
| 2 | 1.0 | 1 | 59.47±15.16 | 88.60±6.96 | 63.51±14.63 | 88.34±7.05 |
| | | 4 | 41.08±13.42 | 83.87±10.92 | 55.43±15.76 | 88.80±8.57 |
| | | 10 | 39.47±13.44 | 82.41±10.79 | 48.52±13.47 | 86.33±8.93 |
| | 0.1 | 1 | 40.48±14.59 | 81.43±12.74 | 57.22±14.05 | 86.81±8.31 |
| | | 4 | 26.72±14.54 | 78.56±12.58 | 44.39±16.49 | 83.12±9.68 |
| | | 10 | 22.29±16.39 | 73.54±10.97 | 36.57±17.23 | 84.37±10.52 |

Table 13: Personalized accuracy comparison on CIFAR10 under various settings with 100 clients. The used network is 4convNet.

| FL settings | | | Personalized accuracy | | | | | | | |
|---|---|---|---|---|---|---|---|---|---|---|
| $s$ | $f$ | $\tau$ | FedBABU (Ours) | FedAvg | FedPer | LG-FedAvg | FedRep | Per-FedAvg | Ditto | Local-Only |
| 10 | 1.0 | 1 | **80.52**±5.51 | 79.72±5.24 | 78.51±5.66 | 80.37±5.32 | 71.96±6.62 | 79.89±5.84 | 79.89±6.49 | 55.40±11.78 |
| | | 4 | **79.30**±5.67 | 73.35±6.15 | 71.04±7.50 | 74.75±5.87 | 63.71±7.01 | 72.10±6.57 | 77.99±6.48 | |
| | | 10 | **76.86**±5.08 | 71.03±5.86 | 69.25±7.07 | 71.91±5.72 | 60.27±7.13 | 66.52±7.64 | 72.88±6.11 | |
| | 0.1 | 1 | **78.80**±5.51 | 73.05±5.94 | 77.22±5.57 | 67.95±7.00 | 78.21±5.38 | 77.23±5.48 | 75.27±6.64 | |
| | | 4 | **77.45**±6.23 | 71.06±6.19 | 71.13±6.78 | 64.17±7.29 | 68.78±6.91 | 67.32±7.99 | 71.49±6.34 | |
| | | 10 | **76.25**±6.16 | 69.11±5.70 | 66.43±6.59 | 61.15±7.32 | 60.70±8.76 | 61.22±8.14 | 64.97±7.07 | |
| 5 | 1.0 | 1 | **85.17**±6.43 | 83.82±6.75 | 84.50±5.70 | 85.04±6.35 | 78.98±8.29 | 74.96±7.36 | 82.64±6.85 | 68.79±13.38 |
| | | 4 | 83.65±5.99 | 77.47±8.32 | 76.96±8.65 | 77.54±8.14 | 73.05±9.11 | 65.17±10.71 | **83.77**±6.93 | |
| | | 10 | **83.48**±6.13 | 77.01±7.80 | 76.17±8.11 | 77.28±7.52 | 70.62±7.36 | 53.31±12.12 | 79.01±7.56 | |
| | 0.1 | 1 | 82.76±6.47 | 75.08±9.34 | 80.69±7.63 | 63.30±10.62 | **83.71**±7.05 | 72.15±10.22 | 72.08±9.92 | |
| | | 4 | **81.31**±7.17 | 74.87±9.64 | 74.74±8.04 | 59.17±10.18 | 77.64±7.51 | 58.32±11.28 | 66.36±9.53 | |
| | | 10 | **80.99**±7.64 | 74.29±8.29 | 74.92±8.32 | 55.19±13.91 | 68.71±10.47 | 47.48±13.76 | 58.66±12.51 | |
| 2 | 1.0 | 1 | 91.41±6.50 | 91.40±6.12 | 91.75±6.90 | **92.35**±5.44 | 91.11±8.21 | 72.02±14.48 | 91.41±6.82 | 90.85±9.10 |
| | | 4 | 91.24±7.33 | 88.91±9.06 | 89.14±8.44 | 88.67±9.03 | 87.74±8.08 | 43.41±23.51 | **92.17**±6.42 | |
| | | 10 | **90.53**±7.20 | 88.48±8.44 | 88.15±9.20 | 87.38±8.56 | 86.19±9.50 | 36.42±18.35 | 90.15±7.57 | |
| | 0.1 | 1 | 90.98±6.22 | 86.36±10.57 | 90.33±7.52 | 68.63±14.34 | **91.88**±7.37 | 63.28±13.77 | 64.16±18.12 | |
| | | 4 | **87.85**±8.57 | 85.94±10.72 | 86.84±10.20 | 47.00±24.75 | 86.68±10.13 | 32.19±19.63 | 54.69±17.67 | |
| | | 10 | **88.93**±9.33 | 85.14±10.73 | 86.69±8.04 | 33.30±22.29 | 83.85±11.20 | 26.61±19.52 | 43.64±21.03 | |

## F.3 ABLATION STUDY ACCORDING TO THE LOCAL UPDATE PARTS

We decouple the entire network in more detail during local updates to investigate whether the body should be totally trained. For this ablation study, we used 4convNet on CIFAR10. Table 14 and Table 15 show the initial and personalized accuracy according to the local update parts, respectively. For consistency, all cases are personalized by fine-tuning the entire network in this experiment. In all cases, FedBABU has the best performance, which implies that the body must be totally trained at least when using 4convNet. It is believed that the same (body-totally) or similar (body except for a few top layers) trends appear when large networks are used.

Table 14: Initial accuracy comparison on CIFAR10 under various settings with 100 clients. The used network is 4convNet.

| FL settings | | | Local update parts | | | | |
|---|---|---|---|---|---|---|---|
| $s$ | $f$ | $\tau$ | Conv1 | Conv12 | Conv123 | Conv1234 (FedBABU) | Conv1234+Linear (FedAvg) |
| 10 | 1.0 | 1 | 25.22±8.34 | 47.23±6.91 | 68.44±6.65 | 71.50±7.35 | 68.89±6.35 |
| | | 4 | 23.02±9.57 | 45.73±7.55 | 60.25±7.69 | 68.40±6.67 | 61.56±6.84 |
| | | 10 | 17.80±8.15 | 42.51±7.03 | 60.48±6.96 | 64.66±6.00 | 60.06±5.68 |
| | 0.1 | 1 | 19.33±7.71 | 42.84±8.15 | 62.71±7.67 | 69.36±5.61 | 63.79±7.59 |
| | | 4 | 19.85±8.96 | 41.58±8.37 | 55.26±6.65 | 66.80±7.43 | 59.65±8.03 |
| | | 10 | 18.99±9.10 | 44.38±7.49 | 54.61±6.66 | 62.68±7.06 | 57.32±6.76 |
| 5 | 1.0 | 1 | 23.42±8.35 | 44.35±9.59 | 64.27±10.48 | 68.54±8.45 | 66.56±9.39 |
| | | 4 | 22.34±9.48 | 40.07±8.83 | 54.50±9.21 | 63.79±9.12 | 52.60±7.79 |
| | | 10 | 18.94±10.38 | 39.59±9.03 | 51.25±9.22 | 61.10±9.80 | 51.55±8.23 |
| | 0.1 | 1 | 20.78±7.96 | 38.90±10.02 | 52.14±8.95 | 63.44±9.82 | 55.13±8.57 |
| | | 4 | 19.33±10.59 | 40.71±9.19 | 47.66±11.17 | 59.93±8.32 | 48.77±8.54 |
| | | 10 | 23.71±8.86 | 36.74±9.72 | 46.97±9.81 | 55.55±9.90 | 46.32±10.52 |
| 2 | 1.0 | 1 | 18.72±16.35 | 45.01±13.57 | 58.86±15.93 | 63.51±14.63 | 59.47±15.16 |
| | | 4 | 20.08±10.09 | 34.58±11.84 | 40.56±12.73 | 55.43±15.76 | 41.08±13.42 |
| | | 10 | 18.81±14.56 | 29.74±10.94 | 37.75±12.08 | 48.52±13.47 | 39.47±13.44 |
| | 0.1 | 1 | 16.17±18.73 | 37.08±12.45 | 45.57±13.36 | 57.22±14.05 | 40.48±14.59 |
| | | 4 | 14.09±18.82 | 28.75±14.13 | 25.12±15.63 | 44.39±16.49 | 26.72±14.54 |
| | | 10 | 17.40±10.53 | 23.73±17.87 | 23.16±19.27 | 36.57±17.23 | 22.29±16.39 |

Table 15: Personalized accuracy comparison on CIFAR10 under various settings with 100 clients. The used network is 4convNet.

| FL settings | | | Local update parts | | | | |
|---|---|---|---|---|---|---|---|
| $s$ | $f$ | $\tau$ | Conv1 | Conv12 | Conv123 | Conv1234 (FedBABU) | Conv1234+Linear (FedAvg) |
| 10 | 1.0 | 1 | 55.94±7.28 | 64.88±8.34 | 78.74±5.24 | 80.52±5.51 | 79.72±5.24 |
| | | 4 | 53.06±7.17 | 62.84±6.57 | 71.85±6.11 | 79.30±5.67 | 73.35±6.15 |
| | | 10 | 52.89±7.14 | 62.02±6.48 | 71.16±6.24 | 76.86±5.08 | 71.03±5.86 |
| | 0.1 | 1 | 53.85±8.04 | 64.40±6.47 | 72.79±6.46 | 78.80±5.51 | 73.05±5.94 |
| | | 4 | 50.75±8.21 | 61.79±6.84 | 66.45±7.31 | 77.45±6.23 | 71.06±6.19 |
| | | 10 | 51.95±7.45 | 62.72±7.64 | 67.41±6.31 | 76.25±6.16 | 69.11±5.70 |
| 5 | 1.0 | 1 | 67.49±9.74 | 74.39±7.74 | 82.55±6.50 | 85.17±6.43 | 83.82±6.75 |
| | | 4 | 66.12±10.02 | 71.77±8.48 | 79.68±6.77 | 83.65±5.99 | 77.47±8.32 |
| | | 10 | 64.87±8.21 | 69.73±9.10 | 76.25±8.51 | 83.48±6.13 | 77.01±7.80 |
| | 0.1 | 1 | 65.50±9.00 | 74.50±8.77 | 74.50±8.77 | 82.76±6.47 | 75.08±9.34 |
| | | 4 | 65.05±8.56 | 70.80±9.29 | 75.44±8.06 | 81.31±7.17 | 74.87±9.64 |
| | | 10 | 65.61±8.73 | 69.66±8.68 | 74.79±7.85 | 80.99±7.64 | 74.29±8.29 |
| 2 | 1.0 | 1 | 85.44±9.20 | 88.20±8.86 | 90.61±6.48 | 91.41±6.50 | 91.40±6.12 |
| | | 4 | 84.46±9.30 | 83.94±10.18 | 88.86±8.62 | 91.24±7.33 | 88.91±9.06 |
| | | 10 | 81.62±9.66 | 86.20±10.72 | 88.58±9.01 | 90.53±7.20 | 88.48±8.44 |
| | 0.1 | 1 | 85.46±9.68 | 85.78±9.73 | 88.18±8.41 | 90.98±6.22 | 86.36±10.57 |
| | | 4 | 82.74±10.45 | 85.13±9.77 | 85.91±10.87 | 87.85±8.57 | 85.94±10.72 |
| | | 10 | 82.52±10.02 | 84.98±9.25 | 85.46±9.74 | 88.93±9.33 | 85.14±10.73 |

# G  RESULTS OF RESNET18 AND RESNET50 ON CIFAR100

## G.1  ACCURACY CURVES IN THE CENTRALIZED SETTING

We experiment with ResNet18 and ResNet50. In the centralized setting, test accuracy curves according to the update part (Figure 6 and Figure 8) and according to the initialization method (Figure 7 and Figure 9) have the same trend as we have seen before.

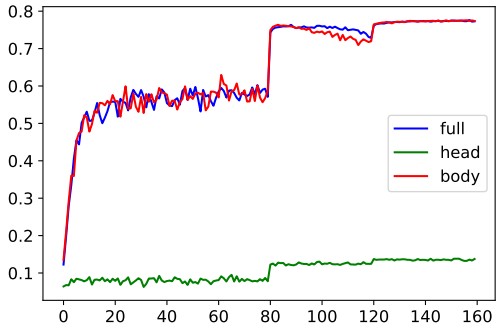

Figure 6: Test accuracy curves of ResNet18 on CIFAR100 according to the update part in the centralized setting.

Figure 7: Test accuracy curves of ResNet18 on CIFAR100 according to the initialization method when the body is only trained.

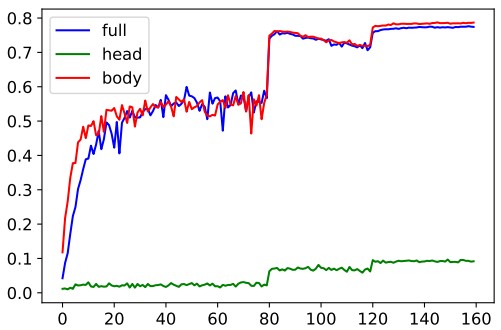

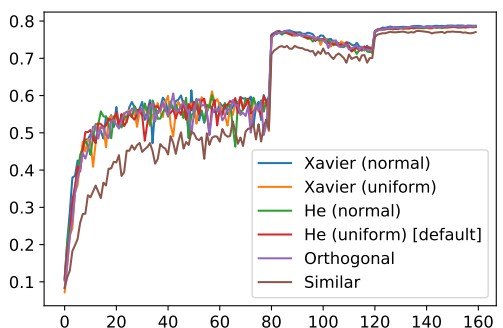

Figure 8: Test accuracy curves of ResNet50 on CIFAR100 according to the update part in the centralized setting.

Figure 9: Test accuracy curves of ResNet50 on CIFAR100 according to the initialization method when the body is only trained.

## G.2 EXPERIMENTAL RESULTS

Table 16 and Table 17 are the results of ResNet18 and ResNet50, respectively. It is shown that the performance gap between FedAvg and FedBABU increases when ResNet is used rather than when MobileNet (in the main paper) is used. Furthermore, it is observed that as the complexity of the model increases (ResNet18 → ResNet50), the performance of FedAvg decreases, while that of FedBABU does not.

Table 16: Initial and personalized accuracy of FedAvg and FedBABU on CIFAR100 under various settings with 100 clients. The used network is ResNet18. $f$ is 0.1.

| FL settings | | Initial accuracy | | Personalized accuracy | |
|---|---|---|---|---|---|
| $s$ | $\tau$ | FedAvg | FedBABU | FedAvg | FedBABU |
| 100 | 1 | 57.26±4.71 | 59.87±4.27 | 60.39±5.16 | 66.48±4.43 |
| | 4 | 44.79±4.70 | 49.02±5.01 | 49.32±4.90 | 55.65±4.81 |
| | 10 | 34.52±4.73 | 40.60±5.62 | 39.04±4.77 | 47.42±5.66 |
| 50 | 1 | 53.73±4.97 | 59.23±5.10 | 63.21±5.33 | 71.16±4.75 |
| | 4 | 42.47±5.21 | 49.03±4.56 | 52.76±5.99 | 62.54±4.92 |
| | 10 | 37.09±4.20 | 40.71±4.98 | 47.13±4.47 | 54.99±5.37 |
| 10 | 1 | 42.88±8.08 | 53.65±7.53 | 76.84±7.08 | 83.83±5.22 |
| | 4 | 29.15±8.18 | 39.76±8.23 | 67.29±6.42 | 77.40±6.45 |
| | 10 | 26.54±7.66 | 30.96±8.09 | 66.18±7.29 | 72.49±6.19 |

Table 17: Initial and personalized accuracy of FedAvg and FedBABU on CIFAR100 under various settings with 100 clients. The used network is ResNet50. $f$ is 0.1.

| FL settings | | Initial accuracy | | Personalized accuracy | |
|---|---|---|---|---|---|
| $s$ | $\tau$ | FedAvg | FedBABU | FedAvg | FedBABU |
| 100 | 1 | 42.89±5.34 | 60.78±4.74 | 47.59±5.16 | 65.74±4.72 |
| | 4 | 33.59±4.35 | 50.71±4.37 | 39.56±4.43 | 56.35±4.43 |
| | 10 | 32.88±4.23 | 40.59±4.65 | 37.32±4.47 | 45.52±5.29 |
| 50 | 1 | 42.14±5.27 | 59.51±4.69 | 53.04±5.28 | 70.58±5.06 |
| | 4 | 30.97±4.43 | 48.65±4.94 | 42.76±4.95 | 59.64±5.74 |
| | 10 | 30.90±4.76 | 41.14±5.47 | 43.28±5.49 | 52.13±5.14 |
| 10 | 1 | 28.18±7.99 | 51.17±6.97 | 65.50±6.92 | 81.43±6.20 |
| | 4 | 19.31±7.56 | 38.45±9.50 | 58.92±7.40 | 73.86±6.96 |
| | 10 | 18.41±7.42 | 30.22±8.26 | 59.50±7.30 | 68.42±8.04 |

# H  RESULTS OF RESNET10 ON DIRICHLET DISTRIBUTION-BASED CIFAR100

We evaluate ours and existing algorithms under unbalanced and non-IID settings derived from the Dirichlet distribution. Figure 10 and 11 describe the distributions of train and test data points and non-IID depending on $\beta$, which is the hyperparameter of the Dirichlet distribution. Table 18 describes the results on Dirichlet distribution-based non-IID CIFAR100 in the realistic FL setting ($f$=0.1 and $\tau$=10). The lower $\beta$ implies the larger heterogeneity.

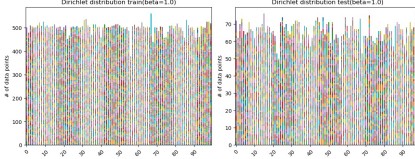
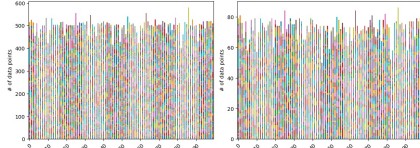

Figure 10: Data distribution ($\beta = 1.0$).       Figure 11: Data distribution ($\beta = 0.5$).

Table 18: Personalized accuracy comparison on Dirichlet distribution-based non-IID CIFAR100 with 100 clients (FL setting: $f$=0.1, and $\tau$=10). The used network is MobileNet.

| $\beta$ | Personalized accuracy | | | | | | | |
|---|---|---|---|---|---|---|---|---|
| | FedBABU (Ours) | FedAvg | FedPer | LG-FedAvg | FedRep | Per-FedAvg | Ditto | Local-only |
| 1.0 | **34.92**±6.22 | 32.10±6.15 | 34.21±5.68 | 25.00±6.12 | 12.68±5.27 | 18.85±4.97 | 7.11±4.84 | 23.73±5.71 |
| 0.5 | **44.67**±5.80 | 40.21±5.68 | 35.62±6.17 | 27.24±6.73 | 12.52±4.32 | 18.36±5.01 | 8.13±4.69 | 31.24±5.49 |

# I  RESULTS OF 3CONVNET ON EMNIST

Table 19 describes the results on EMNIST using a network that consists of three convolution layers and a linear classifier. We set the number of users to 1488, and the number of data points per user was approximately 450. We fixed the fraction ratio at 0.1 and local epochs at 10 (which is the realistic setting used in our paper). The results demonstrate that FedBABU also has the best performance on EMNIST.

Table 19: Personalized accuracy comparison on EMNIST with 1488 clients (FL setting: $f$=0.1, and $\tau$=10). The used network is 3convNet.

| $s$ | Personalized accuracy | | | | | | | |
|---|---|---|---|---|---|---|---|---|
| | FedBABU (Ours) | FedAvg | FedPer | LG-FedAvg | FedRep | Per-FedAvg | Ditto | Local-Only |
| 60 | **85.27**±5.33 | 84.52±4.62 | 54.68±6.77 | 82.75±4.78 | 75.28±5.21 | 78.45±5.22 | 83.38±5.46 | 72.50±5.91 |
| 30 | **89.30**±5.03 | 85.46±5.16 | 58.72±8.04 | 81.64±5.80 | 82.03±5.34 | 80.61±6.02 | 83.23±6.06 | 81.63±5.73 |
| 6 | 95.96±5.00 | 89.83±7.72 | 64.86±15.09 | 73.50±13.02 | 94.61±5.11 | 62.94±14.86 | 76.27±12.72 | **96.06**±3.89 |

## J  RESULTS ON IN-DISTRIBUTION (ID) AND OUT-OF-DISTRIBUTION (OOD) CLASSES

To investigate the accuracy of in-distribution (ID) and out-of-distribution (OOD) classes, we scatter IID-like test set ($s$=100) to all clients. Namely, some classes are in test data set ($s$=100), but not in training data set ($s$=50 or $s$=10). Table 20 describes the results of FedAvg and FedBABU on this experiment. Here, in-distribution accuracy is the test accuracy of the classes present in the training data set; otherwise, out-of-distribution accuracy. Before personalization (Before in tables), FedBABU beats FedAvg from both ID and OOD. After personalization (After in tables), FedBABU becomes fit strongly for the ID classes than OOD classes. The OOD accuracy of FedBABU decreases more than that of FedAvg, but the accuracy of FedAvg simiarly drops after personalization. These results indicate that no personalization is the best strategy for both FedBABU and FedAvg. Therefore, because the OOD accuracy of FedBABU is higher than that of FedAvg before personalization, it is concluded that FedBABU has better performance than FedAvg when the optimal strategy for OOD is used.

Table 20: In-distribution (ID) and out-of-distribution (OOD) accuracy of FedAvg and FedBABU before/after personalization under various settings with 100 clients. The used network is MobileNet.

| FL settings | | | FedAvg | | | | FedBABU | | | |
|---|---|---|---|---|---|---|---|---|---|---|
| | | | ID accuracy | | OOD accuracy | | ID accuracy | | OOD accuracy | |
| $s$ | $f$ | $\tau$ | Before | After | Before | After | Before | After | Before | After |
| 50 | 1.0 | 1 | 47.30±7.05 | 55.14±7.57 | 46.46±6.74 | 32.85±6.67 | 48.30±7.75 | 57.92±7.98 | 47.81±6.56 | 24.06±5.27 |
| | | 4 | 37.30±8.56 | 45.30±8.61 | 37.24±6.29 | 27.05±5.71 | 38.99±7.73 | 49.70±8.00 | 37.76±5.71 | 13.88±4.78 |
| | | 10 | 30.98±7.57 | 38.46±9.04 | 30.54±5.98 | 20.14±5.18 | 27.63±7.41 | 37.40±8.06 | 29.69±6.24 | 5.72±3.32 |
| | 0.1 | 1 | 39.90±7.55 | 47.06±7.96 | 39.06±6.17 | 28.32±5.76 | 42.98±8.20 | 53.35±7.73 | 42.14±7.49 | 16.04±5.08 |
| | | 4 | 33.23±7.88 | 41.25±8.19 | 35.70±6.15 | 27.38±5.63 | 37.07±8.02 | 47.04±9.18 | 36.34±6.10 | 11.67±4.32 |
| | | 10 | 28.70±7.77 | 36.32±7.41 | 27.15±6.00 | 18.05±5.44 | 28.26±8.44 | 38.74±9.08 | 29.06±5.61 | 6.70±2.94 |
| 10 | 1.0 | 1 | 42.16±18.36 | 78.14±16.62 | 41.51±5.12 | 14.25±3.37 | 45.76±16.41 | 79.60±11.35 | 44.93±5.13 | 3.93±2.24 |
| | | 4 | 30.60±15.77 | 68.38±16.14 | 31.43±5.02 | 7.13±3.10 | 38.43±17.27 | 71.59±15.29 | 37.05±5.19 | 0.13±0.36 |
| | | 10 | 26.30±16.40 | 61.21±17.16 | 23.70±4.50 | 2.27±1.48 | 27.20±15.12 | 64.21±16.24 | 25.49±4.42 | 0.00±0.00 |
| | 0.1 | 1 | 34.47±18.00 | 69.23±14.87 | 34.10±4.87 | 9.74±2.95 | 42.57±19.11 | 75.18±16.46 | 39.64±5.04 | 3.32±1.89 |
| | | 4 | 26.86±16.09 | 64.24±16.45 | 27.09±4.55 | 6.03±2.73 | 31.31±17.27 | 68.38±15.55 | 30.75±5.23 | 0.06±0.24 |
| | | 10 | 18.42±12.83 | 56.21±17.46 | 19.67±3.89 | 2.95±1.92 | 22.30±14.65 | 64.73±17.16 | 22.46±4.71 | 0.00±0.00 |

## K  SIMILARITY BETWEEN CLIENTS DURING FINE-TUNING

We investigate the cosine similarity between different clients during 20 fine-tuning epochs, described in Figure 12. During fine-tuning for personalization evaluation, the entire network is updated in the cases of both FedAvg and FedBABU. From the results, it is found that classifier (black line in Figure 12) is closely related to personalization. In other words, all clients have similar extractors during fine-tuning (i.e., large cosine similarity), while make different classifiers based on their own data. Furthermore, it is observed that the cosine similarity on the head between different clients

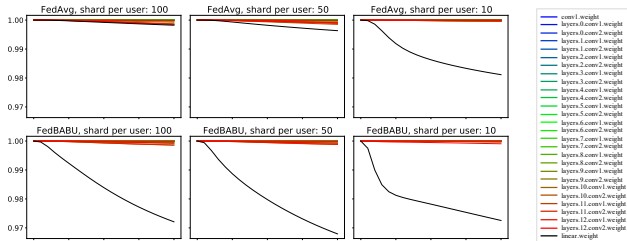

Figure 12: Layer-wise cosine similarity of FedAvg and FedBABU trained under the realistic FL setting ($f$=0.1 and $\tau$=10) during fine-tuning. The blue-, green-, and red-like lines represent low-, middle-, and high-level convolution layers, respectively. The black line represents the last linear layer (i.e., head).

decreases more rapidly in the case of FedBABU than FedAvg. It is believed that this is because the classifier of FedAvg starts personalization from the federated convergent classifier, while that of FedBABU starts personalization from the initialized orthogonal classifier. This characteristic explains rapid personalization, introduced in Section 5.2.4.

## L  FedBABU with Body Update on the Server

Table 21: Initial and personalized accuracy of FedBABU on CIFAR100 under realistic FL settings ($N$=100, $f$=0.1, and $\tau$=10) according to the $p$, which is the percentage of all client data that the server also has. Here, the body is updated only on the server using available data.

| $p$ | $s$=100 (heterogeneity ↓) | | $s$=50 | | $s$=10 (heterogeneity ↑) | |
|---|---|---|---|---|---|---|
| | Initial | Personalized | Initial | Personalized | Initial | Personalized |
| 0.00 | 29.38±4.74 | 35.94±5.06 | 27.91±5.27 | 42.63±5.59 | 18.50±7.82 | 66.32±7.02 |
| 0.05 | 28.68±4.65 | 36.25±4.77 | 26.94±4.98 | 41.01±5.34 | 21.32±5.85 | 66.56±7.24 |
| 0.10 | 32.49±4.52 | 39.93±4.76 | 31.39±4.84 | 47.02±5.54 | 24.34±5.32 | 68.95±6.92 |

Let us repeat the experiment in Section 4 again, where the server has a small portion $p$ of the non-privacy data of the clients. Table 21 describes the initial and personalized accuracy of FedBABU when the global model is body-partially updated on the server using available data (such as experiments (B) in Table 2). FedBABU with body updates on the server improves personalization compared to FedAvg with body updates on the server, maintaining the initial accuracy (refer to (B) in Table 2). This result demonstrates that training the head negatively affects personalization even when trained locally. In addition, the performance of FedBABU improves as $p$ increases. This implies that there is room for enhancing the representation beyond that of FedBABU.

## M  Body Aggregation and Body Update on the FedProx

Even when all clients are active every communication round (i.e., $f$=1.0), the personalized performance of FedProx+BABU beats not only that of FedAvg but also that of FedBABU.

Table 22: Initial and personalized accuracy of FedProx and FedProx+BABU with $\mu$ of 0.01 on CIFAR100 with 100 clients and $f$ of 1.0.

| Algorithm | $\tau$ | $s$=100 (heterogeneity ↓) | | $s$=50 | | $s$=10 (heterogeneity ↑) | |
|---|---|---|---|---|---|---|---|
| | | Initial | Personalized | Initial | Personalized | Initial | Personalized |
| FedProx | 1 | 42.25±4.58 | 56.22±4.54 | 52.08±4.92 | 59.95±4.99 | 44.39±7.53 | 78.96±6.16 |
| | 4 | 40.32±4.70 | 43.17±4.62 | 40.11±5.80 | 45.99±6.05 | 29.73±6.74 | 67.66±7.67 |
| | 10 | 30.75±4.53 | 33.43±4.48 | 29.44±4.24 | 35.65±4.60 | 19.63±5.73 | 57.83±7.13 |
| FedProx +BABU | 1 | 55.02±4.42 | 61.48±4.83 | 54.13±4.86 | 66.54±4.68 | 50.42±9.63 | 83.75±5.97 |
| | 4 | 39.01±4.97 | 46.31±4.98 | 38.79±5.15 | 52.96±5.46 | 34.09±7.28 | 77.28±5.96 |
| | 10 | 28.69±4.64 | 36.29±4.45 | 30.39±5.39 | 44.46±5.49 | 25.16±5.84 | 69.77±6.26 |

## N  Discussion on the effectiveness of FedAvg

MOCHA (Smith et al., 2017) is a representative personalized FL paper using regularization based on relationships between tasks. pFedMe (T Dinh et al., 2020) and Ditto (Li et al., 2021) train local models with a regularization based on the divergence between a global model and local models. If the weight on regularization is set to zero, these regularized personalized FL algorithms reduce to the local-only algorithm; on the other extreme, these reduce to the FedAvg algorithm (i.e., $\theta_1 = \cdots = \theta_N$). The personalized FL algorithms such as FedPer (Collins et al., 2021), in which each client has parts that are not aggregated (i.e., personalized parts), can be explained similarly. If there is no personalized part, these personalized FL algorithms reduce to the FedAvg; on the other extreme, i.e., if the entire network is not aggregated, these personalized FL algorithms reduce to the local-only algorithm. In summary, personalized FL algorithms are in-between local-only and FedAvg algorithms and have better personalized performance than both local-only and FedAvg in general.

In the same vein, FedAvg+Fine-tuning is also in-between local-only and FedAvg because this algorithm uses two extreme algorithms sequentially (i.e., FedAvg followed by local-only). However, intuitively, FedAvg+Fine-tuning is a bit more FedAvg-based, whereas personalized FL algorithms

are a bit more local-only-based. This is because FedAvg+Fine-tuning is the same as FedAvg before personalization, whereas personalized FL algorithms develop different clients' models from scratch like local-only. Therefore, we believe that *FedAvg+Fine-tuning earns more benefits (particularly for representation) from federation* than personalized FL algorithms from this difference (FedAvg-based v.s. local-only-based). An additional difference between FedAvg+Fine-tuning and regularized personalized FL algorithms is that FedAvg+Fine-tuning optimizes clients *separately* using their own data set based on the well-federated extractor, whereas regularized personalized FL algorithms optimize clients *jointly* from scratch. In other words, FedAvg+Fine-tuning can easily optimize all clients' models based on their own data sets. However, it is too difficult to optimize all clients' models simultaneously. Therefore, FedAvg+Fine-tuning can achieve better personalized performance than regularized personalized FL algorithms.

Furthermore, very recent work (Cheng et al., 2021) has addressed the effectiveness of FedAvg+Fine-tuning theoretically. In Cheng et al. (2021), they compared local-only (zero collaboration), FedAvg (zero personalization) (McMahan et al., 2017), FedAvg+Fine-tuning, Per-FedAvg (Fallah et al., 2020), and pFedMe (T Dinh et al., 2020) in the high-dimensional asymptotic limit by analyzing bias-variance. They demonstrated that the asymptotic test loss of FedAvg+Fine-tuning (FTFA in Cheng et al. (2021)) matches that of Per-FedAvg and that the asymptotic test loss of FedAvg+Fine-tuning with a ridge regularizer (RTFA in Cheng et al. (2021)) matches that of pFedMe on their stylized linear regression model.

Although Cheng et al. (2021) thoroughly addressed this problem on their stylized linear regression model, there are few results on real data sets except for test accuracies. It is believed that "why FedAvg+Fine-tuning is effective" itself needs to be studied more using real data sets, as "why MAML (Finn et al., 2017) is effective" has been addressed in (Raghu et al., 2019).

## O    FEDAVG WITH DIFFERENT LEARNING RATES

Table 23: FedAvg with different learning rates under realistic FL setting ($f$=0.1 and $\tau$=10). We set the body's initial learning rate ($\alpha_b$) as 0.1.

| $s$ | FedAvg ($\alpha_h=\alpha_b$) | FedAvg ($\alpha_h=0.1 \times \alpha_b$) | FedAvg ($\alpha_h=0.01 \times \alpha_b$) | FedBABU ($\alpha_h=0$) |
|---|---|---|---|---|
| 100 | 24.34±4.58 | 27.10±4.43 | 28.37±4.60 | 29.36±4.46 |
| 50 | 33.10±5.08 | 35.97±5.17 | 36.21±5.08 | 36.49±5.37 |
| 10 | 50.25±6.27 | 52.96±7.52 | 54.04±7.60 | 54.93±7.85 |

What we want to emphasize is the importance of making all clients have the same class boundary by freezing the head. We argue that this shared classifier enhances the representation power of the federated model, which is an important factor for personalization. From this perspective, if the head moves even a little bit client by client, the different class boundaries to train feature extractor are used during local updates. Therefore, the federated model's feature extractor might hurt.

To verify this hypothesis, we experiment using different learning rates depending on the parts. The above table describes the federated model's accuracy without a classifier (i.e., w/o classifier accuracy in Table 3) to investigate representation power of models. We set the body's initial learning rate as 0.1, the fraction ratio as 0.1, and local epochs as 10. $\alpha_h$ and $\alpha_b$ are the head's learning rate and the body's learning rate, respectively.

From this result, we believe that the federated model's representation power increases as the head's learning rate decreases, i.e., as a learning signal for the body is larger. In this trend, we highlight that the best performance is due to the same classifier on all clients.

## P    CLASS-WISE ANALYSIS DURING FEDERATED TRAINING

To explain why FedBABU has better performance than FedAvg after federated training (i.e., before fine-tuning), we analyze the federated training procedure of FedAvg and FedBABU class-wisely under the realistic federated settings ($f$=0.1 and $\tau$=10). For this analysis, we assume that each client has 500 training data and 10,000 test data (i.e., the whole test data) of CIFAR-100. Test data set is divided into two groups for each client; in-class test data and out-of-class test data. In-class test

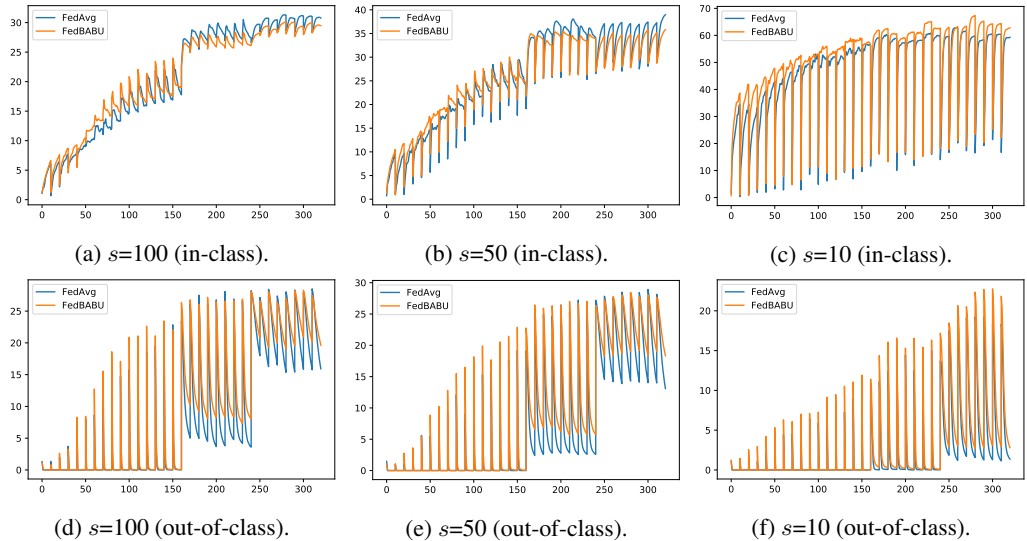

Figure 13: In-class and out-of-class test accuracy curves according to the heterogeneity. The models are trained under the realistic federated settings ($f$=0.1 and $\tau$=10).

data indicates test data whose class is in the classes in training data, whereas out-of-class test data indicates test data whose class is not in the classes in training data. For instance, if client A has classes 0-9 as training data, then in-class test data indicates test data whose class is one of 0-9 and out-of-class test data indicates test data whose class is one of 10-99.

Figure 13 describes the in-class ((a)-(c)) and out-of-class ((d)-(f)) test accuracy curves according to the heterogeneity. Client sampling, broadcasting, local updates, and aggregation stages are repeated during federated training. The accuracies per epoch are averaged right after broadcasting the aggregated model to the selected clients and during local updates on them. It is observed that in-class accuracy of the aggregated model is significantly low and increases dramatically during local updates, whereas out-of-class accuracy of the aggregated model is significantly high and decreases dramatically during local updates, as heterogeneity is larger. In all cases, FedBABU has higher out-of-class accuracy than FedAvg after local updates. It implies that *freezing the head makes less damage to out-of-class during local updates and can lead to better aggregation*. Furthermore, when heterogeneity is large ($s$=10), FedBABU has higher in-class accuracy than FedAvg after local updates.

## Q   FEDBABU WITH THE NON-ORTHOGONAL CLASSIFIERS

We demonstrate that an orthogonal initialization on the head is required for desirable performance in the centralized setting (Appendix B). To investigate whether this characteristic is still required under FL settings, we compare FedBABU with the orthogonal head (which is proposed) to FedBABU without the orthogonal head. FedBABU without the orthogonal head is designed in the same way as 'similar' in Appendix B. Table 24 describes the results. As we expected, if the head does not consist of the orthogonal row vectors, FedBABU cannot achieve desirable performance.

Table 24: Initial and personalized accuracy of FedBABU on CIFAR100 according to the head's orthogonality under various FL settings with 100 clients ($f$=0.1) . MobileNet is used.

| Orthogonal | $\tau$ | $s$=100 (heterogeneity ↓) | | $s$=50 | | $s$=10 (heterogeneity ↑) | |
|---|---|---|---|---|---|---|---|
| | | Initial | Personalized | Initial | Personalized | Initial | Personalized |
| O (proposed) | 1 | 41.02±4.99 | 49.67±4.92 | 41.33±5.10 | 56.69±5.16 | 35.05±7.63 | 76.02±6.29 |
| | 4 | 36.77±4.47 | 44.74±5.10 | 34.68±4.58 | 49.55±5.58 | 25.67±7.31 | 71.00±6.63 |
| | 10 | 29.38±4.74 | 35.94±5.06 | 27.91±5.27 | 42.63±5.59 | 18.50±7.82 | 66.32±7.02 |
| X | 1 | 11.05±3.31 | 17.30±3.41 | 11.01±4.07 | 23.22±4.16 | 9.37±4.73 | 47.09±7.93 |
| | 4 | 18.68±3.70 | 24.13±3.94 | 18.32±4.69 | 29.87±4.51 | 14.02±7.20 | 58.63±8.43 |
| | 10 | 20.46±4.10 | 25.23±4.05 | 20.81±5.11 | 32.82±4.75 | 11.71±6.67 | 58.54±8.67 |

# R    DESCRIPTION OF DATA DISTRIBUTION ACCORDING TO THE SHARDS PER USER $s$

To help understanding data distribution of clients according to the shard per user $s$, we provide examples of them. Figure 14 describes examples of data distribution of clients when $s$ is 10 (left) or 2 (right). Consider that there are 10 clients and CIFAR10 dataset is scattered to each client. When $s$ is 10, total shards 100 because 10 users and 10 shards per user. Then, the CIFAR10 dataset is divided into 100 shards, hence each shard consits of 500 samples with the same class. Here, each client has 10 shards randomly. In this situation, each client can have up to 10 classes. On the contrary, when $s$ is 2, total shards are 20 because 10 users and 2 shards per user. Then, the CIFAR10 dataset is divided into 20 shards, hence each shard consits of 2500 samples with the same class. Here, each client has 2 shards randomly. In this situation, each client can have 2 classes at most. Namely, as $s$ gets smaller, the number of classes that each clients has is limited (i.e., label distribution owned by each client is limited) and the number of samples per class increases.

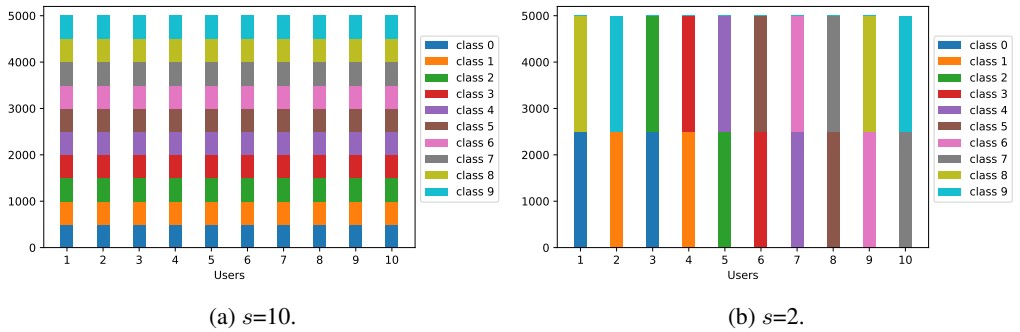

(a) $s$=10.                                                 (b) $s$=2.

Figure 14: Examples of data distribution of users according to the shards per user.

# S    PERFORMANCE WITH LARGER TOTAL EPOCHS

We fixed the total number of epochs to 320, i.e., $K$ (total communication rounds) $\times \tau$ (local epochs) = 320. This is because the performance gap between algorithms appeared and Collins et al. (2021) used total number of epochs to 500, where $K$=100 and $\tau$=5 on CIFAR100. However, because algorithms may not converge, we evaluate them with larger total epochs of 640. Furthermore, we add the results when $s$ is 20 and 5. Table 25 describes the results with the total epochs of 640 when $f$ is 0.1. The results demonstrate that FedBABU is the best under large heterogeneity (i.e., $s$=20, 10, and 5), which is the situation federated learning researchers attempt to solve.

Table 25: Personalized accuracy comparison on CIFAR100 under various settings with 100 clients and MobileNet is used with the total epochs of 640 ($f$=0.1).

| FL settings | | Personalized accuracy | | | | | | |
|---|---|---|---|---|---|---|---|---|
| $s$ | $\tau$ | FedBABU (Ours) | FedAvg (2017) | FedPer (2019) | LG-FedAvg (2020) | FedRep (2021) | Per-FedAvg (2020) | Ditto (2021) |
| 100 | 1 | **59.90**±4.52 | 54.34±4.83 | 57.34±4.66 | 53.24±4.71 | 24.19±3.88 | 48.41±7.23 | 51.86±4.98 |
| | 4 | 45.65±4.67 | 43.70±4.72 | **47.00**±4.62 | 43.87±4.85 | 18.19±3.86 | 41.79±7.08 | 36.46±4.18 |
| | 10 | 36.88±4.62 | 36.25±4.17 | **39.66**±4.96 | 35.65±4.48 | 14.60±3.41 | 32.27±7.42 | 29.11±4.59 |
| 50 | 1 | **65.15**±5.20 | 57.10±4.69 | 61.67±4.91 | 53.89±4.90 | 32.94±5.10 | 46.11±7.84 | 50.78±5.84 |
| | 4 | 52.66±5.79 | 49.42±5.11 | **52.70**±5.14 | 48.18±5.06 | 25.83±5.06 | 36.51±7.83 | 36.09±5.49 |
| | 10 | **47.00**±5.19 | 40.33±5.32 | **47.00**±5.36 | 38.31±5.62 | 21.59±4.18 | 30.31±8.10 | 30.08±5.73 |
| 20 | 1 | **75.12**±5.33 | 66.87±5.16 | 71.51±5.53 | 59.82±5.63 | 50.95±6.17 | 41.77±8.93 | 47.76±8.87 |
| | 4 | **67.71**±5.26 | 63.01±5.50 | 65.87±6.71 | 54.76±6.35 | 42.71±6.04 | 28.71±9.00 | 36.05±10.58 |
| | 10 | **57.55**±6.95 | 52.44±4.73 | 56.47±6.03 | 42.11±8.77 | 37.09±5.75 | 23.39±7.91 | 27.49±8.32 |
| 10 | 1 | **82.26**±5.49 | 76.32±6.22 | 79.44±5.44 | 69.23±6.97 | 66.19±8.16 | 38.64±10.37 | 42.82±14.08 |
| | 4 | **78.67**±6.08 | 73.38±6.42 | 74.00±6.11 | 54.67±10.20 | 57.42±8.14 | 19.81±11.12 | 31.14±15.06 |
| | 10 | **72.72**±6.77 | 68.04±6.81 | 68.41±6.46 | 42.44±12.41 | 48.72±8.30 | 15.39±9.44 | 23.08±14.55 |
| 5 | 1 | **88.16**±5.70 | 84.42±6.43 | 85.04±6.30 | 81.03±6.84 | 77.80±9.18 | 42.10±15.62 | 42.94±19.36 |
| | 4 | **84.91**±7.03 | 80.26±7.35 | 78.65±8.61 | 52.84±17.17 | 69.88±8.43 | 8.36±11.70 | 22.61±21.98 |
| | 10 | **81.05**±6.89 | 74.70±7.50 | 75.19±7.81 | 30.55±19.75 | 62.77±8.05 | 5.92±9.08 | 14.80±21.38 |

## T QUALITATIVE COMPARISON BETWEEN FEDAVG AND FEDBABU

Figure 15 describes t-SNE visualization (Van der Maaten & Hinton, 2008) of representations learned by FedAvg and FedBABU on CIFAR10 and CIFAR100 for a qualitative comparison. For speed acceleration, we used t-SNE-CUDA (Chan et al., 2019). For CIFAR100, the sparse classes are converted into coarse classes (e.g., bicycle → vehicle). However, the difference cannot be captured by the naked eye. Therefore, we further provide t-SNE visualization using only sub-classes of 'vehicle' super-class, described as Figure 16. For CIFAR10, airplane, automobile, ship, and truck classes are used, and for CIFAR100, bicycle, bus, motorcycle, pickup truck, train, lawn-mower, rocket, streetcar, tank, and tractor are used.

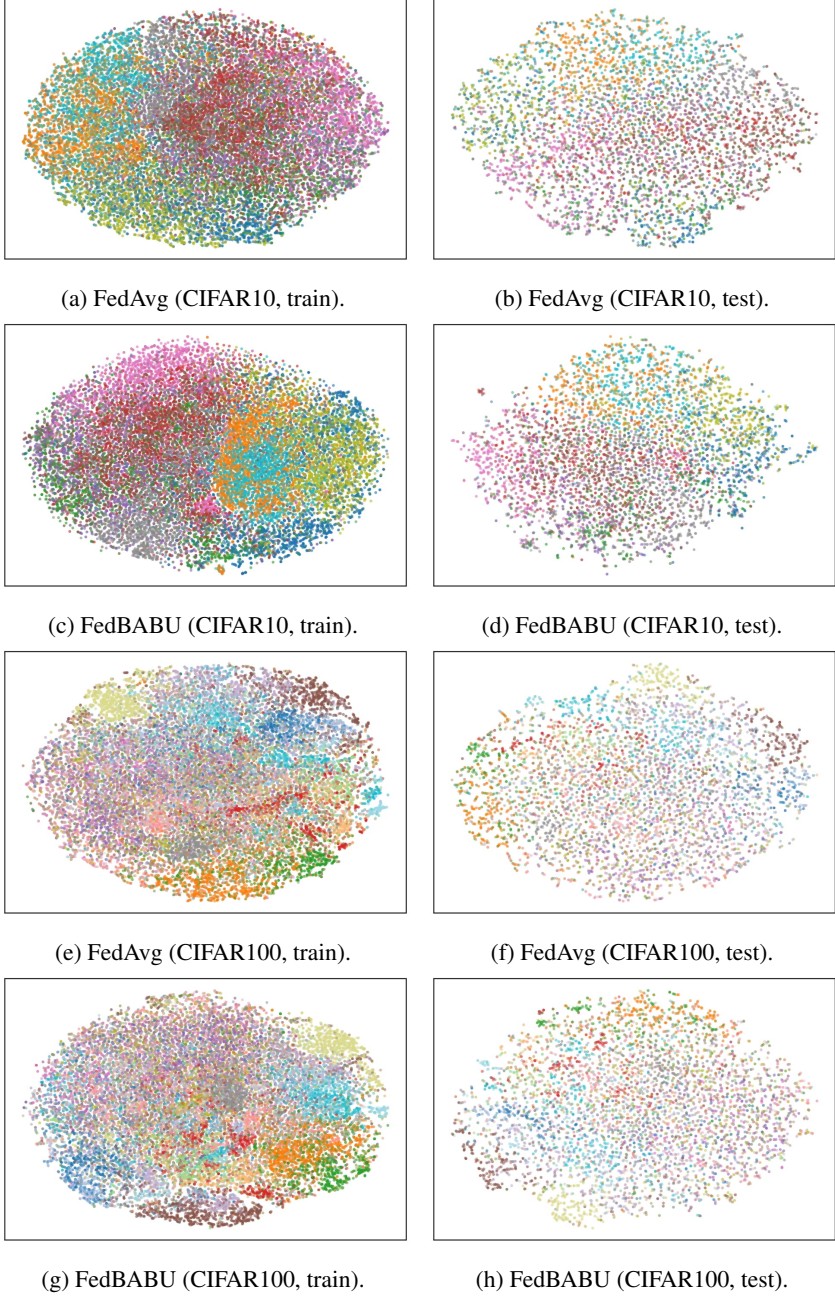

(a) FedAvg (CIFAR10, train).                    (b) FedAvg (CIFAR10, test).

(c) FedBABU (CIFAR10, train).                   (d) FedBABU (CIFAR10, test).

(e) FedAvg (CIFAR100, train).                   (f) FedAvg (CIFAR100, test).

(g) FedBABU (CIFAR100, train).                  (h) FedBABU (CIFAR100, test).

Figure 15: t-SNE visualizations of representations learned by FedAvg and FedBABU. For CIFAR10 and CIFAR100, $s$ is set to 2 and 10, respectively. The models are trained under the realistic federated settings ($f$=0.1 and $\tau$=10).

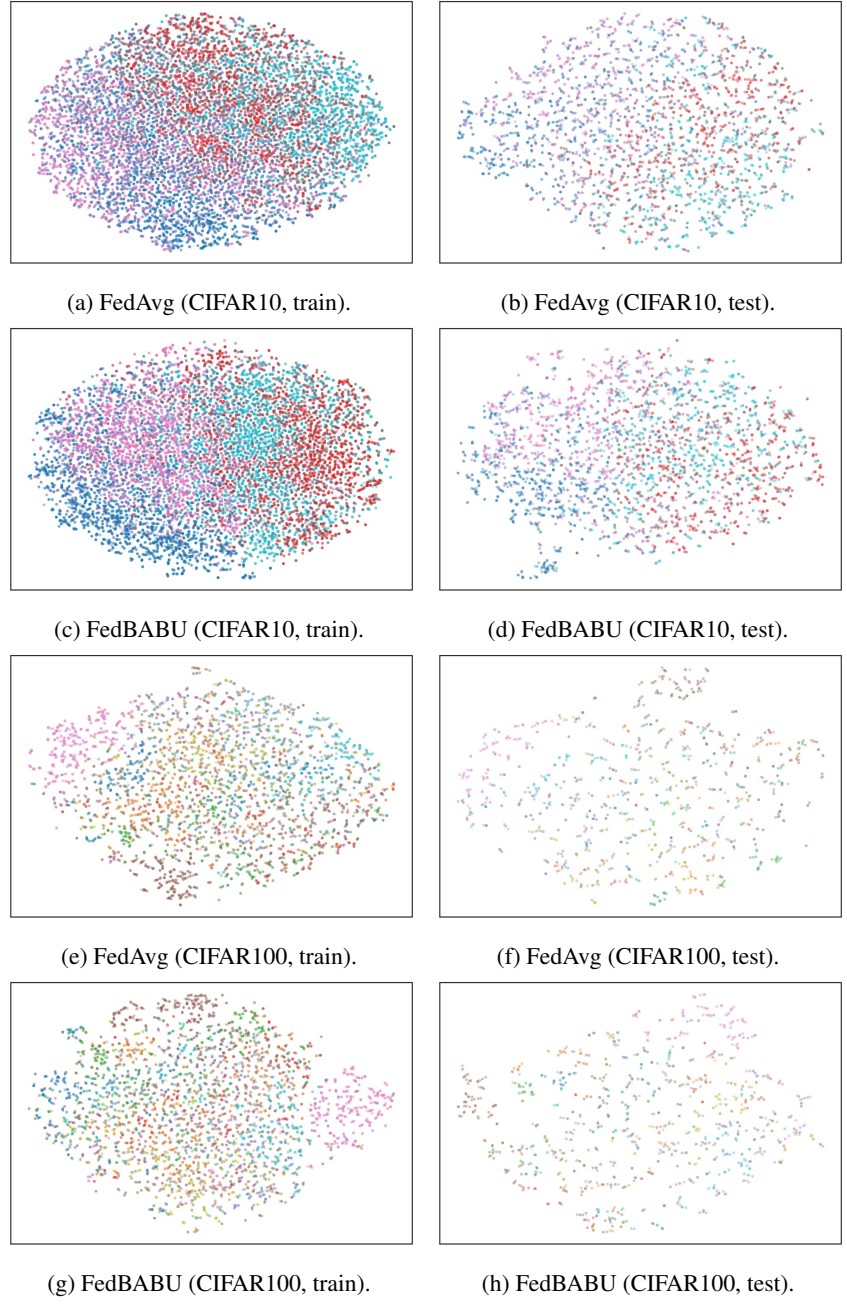

(a) FedAvg (CIFAR10, train).        (b) FedAvg (CIFAR10, test).

(c) FedBABU (CIFAR10, train).        (d) FedBABU (CIFAR10, test).

(e) FedAvg (CIFAR100, train).        (f) FedAvg (CIFAR100, test).

(g) FedBABU (CIFAR100, train).        (h) FedBABU (CIFAR100, test).

Figure 16: t-SNE visualizations of representations learned by FedAvg and FedBABU using only sub-classes of 'vehicle' super-class. For CIFAR10 and CIFAR100, $s$ is set to 2 and 10, respectively. The models are trained under the realistic federated settings ($f$=0.1 and $\tau$=10).

