# OpenReview forum: "FedBABU: Toward Enhanced Representation for Federated Image Classification"
_ICLR.cc/2022/Conference — ICLR 2022 Poster_

### Official Review · Reviewer_UZKM · 2021-10-28

**Correctness:** 3
**Technical Novelty And Significance:** 2
**Empirical Novelty And Significance:** 3
**Recommendation:** 5
**Confidence:** 3

**Main Review:**

Overall, I feel that the paper is promising, but empirically underpowered to support the claims it is making. The overall concept for FedBABU aligns with broader empirical trends in the ML community related to transfer learning, but this does not eliminate the authors' burden to provide either strong empirical results, or theoretical motivation (or both) for their work. As-is, the claims made in the paper are too strong given that all experiments are only on one dataset which is synthetically partitioned into clients. I think that it could also use some revisions for clarity, both in writing and presentation of results. I provide other detailed feedback below.

**Summary Of The Paper:**

This paper presents a new personalization algorithm for FL wherein only the "body" of a model is updated during federated training (not the terminal linear layer), and then a model is fine-tuned (or "personalized") on client data. The authors suggest that this resolves a tension in the existing literature/methods, where better global performance typically comes at the expense of worse local performance. Results are supported with experiments on CIFAR100 with comparisons to several baseline personalized FL methods, and FedAvg.

**Summary Of The Review:**


## Major Comments

* The paper makes some very strong claims which I don't find to be fully supported by the experiments. For example, "we argue that training the head using shared data negatively impacts the personalization". This is a strong, broad claim, but it is made on the basis of experiments on only a single, artifically-federated dataset (CIFAR100). In the absence of (i) stronger empirical evidence, or (ii) some theoretical motivation for this claim (either novel theory or via connections to existing theory), it is difficult to fully get on board with these claims, which form the foundation of the proposed algorithm. If there is empirical evidence from other works supporting the authors' conclusions here, please also cite it. (Another claim needed more support is that training the "head" during federated rounds degrades performance.)

* The claim that "randomly fixed classifiers are acceptable..." is not clearly connected to the authors' statements about orthogonality, which are relegated to the Appendix. Even on reading Appendix B, it's not clear to me how this discussion of initialization proves the authors' claim, or whether this also might imply that personalization of the head might also be unnecessary under certain conditions. Perhaps you could draw a more explicit connection, or link this to work on randomly-featured models, since I sense a connection.

* It's not clear to me why the authors opt to fine-tune the entire model, in order to accommodate FedAVG (mentioned at end of Section 5.2.2), when the results on FedBABU suggest that fine-tuning only the head is sufficient. This seems to require extra compute, and increase the risk of overfitting, while providing no benefit to the algorithm of interest, FedBABU. Please clarify. (I will note that, in T5 (Raffel et al 2020), the authors show that fine-tuning the entire model is preferable to fine-tuning only the later layers, if I remember correctly.)

* I feel that a key analysis missing from the paper is related to the degree of overfitting (or not) that happens during fine-tuning. It is unsurprising that fine-tuning a small amount improves performance, on the clients' data; it is important to know, however, if that results in a much poorer performance on out-of-distribution data (i.e. another client, or a more globally-representative dataset); I was surprised not to see such an analysis in the paper, particularly because this seems of both theoretical importance (the authors imply a kind of tradeoff between global and local model accuracy) and practical importance (it is likely that real-world client data will drift). I would suggest some empirical evidence to this end, or at the very least some discussion of it.

* The finding, in Sec 5.2.3, that FedAVG outperforms *all* preexisting personalization algorithms, except FedBABU, is to me very significant. If true, I think that this should be a "main contribution" of the paper and should be mentioned much earlier, since it suggests that this is the only current personalization algorithm that actually achieves its intended aim.

## Minor Comments

* The tables are very hard to read; I would recommend replacing them with bar graphs (with error bars) and optionally including the tables in the supplement.

* The switching nomenclature between body/extractor and head/classifier was quite confusing to me as a reader, particularly since "classifier" often refers to the *entire* model and I am inclined to read it that way. Please choose one terminology and use it consistently.

* Fig. 1 could be improved a great deal. It needs an illustration of "head" vs. "body", as this is a (the?) major difference from FedAvg that is not currently aptured. Along those lines, it would also be useful to note what is transmitted to the server in the rightmost column, since again, this differs between FedBABU and FedAvg. Also, note that the figure has room to be wider; perhaps you could add a fourth column characterizing the personalization step of FedBABU.

* It seems there is no explicit connection made to the "pretrain -> fine-tune" paradigm that is increasingly prevalent in the broader machine learning community (now also referred to as the use of "foundation models"), although some works in this area are cited (e.g. BERT). An explicit connection would be useful here, and would help motivate the use of a similar approach in FL via FedBABU. Please clarify how, if at all, you believe this method to be different from this transfer learning paradigm.

* On p.5, the claim that global training pays attention to "unnecessary and confusing information for personalization" seems like an incorrect characterization. It may be "irrelevant" to a specific client at a given point in time, but we might reasonably believe that such data actually improves the generalization capabilities of the model (including if that clients' distribution changes in the future) -- does that make it "unnecessary"?


* The authors reference multiple times using a learning rate of zero to not update the head layer (which implies that a no-op is performed, for every paramter, with learning rate zero)...why are these updated being performed at all, instead of only updating \theta_{ext}?

* The conclusion references "regularization" as reducing models' ability to personalize; I am not sure which part of the analysis/experiments this refers to. Please clarify.

## Typos etc.

* All of the "i.e."s in the abstract should be removed.

* P.1: "popular networks have one linear layer...and ResNet" -- this sentence is confusing; please revise.

* P.2 The phrase "representation learning based on the same fixed criteria" is not clear; a similar phrase is used in 5.2. Please clarify what a "criteria on learning representations" (5.2) is and what the "intuition" you are suggesting here may be.

P.3 "When the bottom layers are matched with the body" and "When the top layers are matched with the head" are unclear.

* P4 "decayed 0.1 times" --> decayed by a factor of 0.1

* Why abbreviate (B), (F) in Table 1? I would suggest "Body Updates" vs. "Head + Body Updates" or similar.

## Update after revisions

I thank the authors for their response. Having read the author response, reviewed the updated manuscript, and considered the other reviews, I am keeping my score the same. While the authors addressed several more minor points regarding the paper, I don't believe they addressed my main issue with the paper, which is that the empirical evidence is fairly weak and limited to only a single dataset; I think that either more empirical evidence, or some theoretical support for the proposed method, is needed before being able to argue for acceptance.

---

> ### Author Response · Authors · 2021-11-17
> **Response to Reviewer UZKM**
>
> We itemize the weaknesses or comments you mentioned and answer to them.
>
> # Major
> Q1. More evidence is needed.
> - The decoupling parameters papers we cited, such as (Kang et al., Yu et al.) in Introduction, address the problem of the head in class-imbalanced environments, inspiring our algorithm. Furthermore, very recent work [a], published in NeurIPS 2021, posed a 'classifier bias' problem in federated learning. They and we pose the same problem differently, and they proposed a method to calibrate the head, and we propose a method not to update the head to solve this problem.
> - For evidence, "Kang et al. (2019) and Yu et al. (2020) demonstrated that the head is biased in class-imbalanced environments." is added in Section 1. "Concurrently, Luo et al. (2021) [a] posed a similar problem, the head can be biased because it is closest to the client's label distribution." is added in Section 4.
> - [a] No fear of heterogeneity: Classifier calibration for FL with non-IID data
>
> Q2. The connection randomly fixed classifiers to orthogonality.
> - Our claim "randomly fixed classifiers are acceptable" is explained by the two observations in Appendix B; training only the body with the fixed orthogonal head has desirable performance (i.e., orthogonal fixed classifiers are acceptable) and random initialization on the head is theoretically orthogonal. The former is demonstrated by the decreasing performance when the head has similar row vectors (red line in Figure 3 in Appendix B)
> - Section 5.1 shows that there is no need to train the head if the head is initialized orthogonally (or randomly in high dimension), which is why we can update only the body during local training in the FL settings. (c.f., Section 4 shows the negative impact of training the head and Section 5.1 demonstrates that training the head is not necessarily required.)
>
> Q3. Fine-tuning the entire model is redundant.
> - As you commented, fine-tuning only the head is sufficient for FedBABU, reducing computation costs. It may be a useless concern, but we thought FedAvg would take a bad strategy in terms of performance comparison if the head is fine-tuned only. Therefore, for fair comparison from a performance perspective, we fine-tune models entirely.
> - In addition, although the paper you recommended addresses NLP problems, we thought it is in line with the fact that FedAvg has better personalization performance when the entire network is fine-tuned than when the head is fine-tuned. In this respect, we think FedBABU has one more strength.
>
> Q4. OOD performance during fine-tuning.
> - Appendix J addresses the ID (in-distribution) and OOD (out-of-distribution) performance of FedAvg and FedBABU before and after fine-tuning. Through fine-tuning, FedBABU becomes fit strongly for the ID classes rather than OOD classes. The OOD accuracy of FedBABU decreases more than that of FedAvg, but the accuracy of FedAvg similarly drops after personalization. These results indicate that no personalization is the best strategy for both FedBABU and FedAvg for OOD. Therefore, because the OOD accuracy of FedBABU is higher than that of FedAvg before fine-tuning, it is concluded that FedBABU has better performance than FedAvg when the optimal strategy for OOD is used.
>
> Q5. Superior performance of FedAvg.
> - The third bullet of our contributions is modified, reflecting your opinion. We were also surprised by the performance of FedAvg. The similar results were reported in the very recent paper (Please refer to footnote 8 on page 8) and they said that this effectiveness should be studied deeply. We provide further discussion on the effectiveness of FedAvg in Appendix N.

---

> ### Author Response · Authors · 2021-11-17
> **Response to Reviewer UZKM**
>
> # Minor
> Q1. Readability of the tables.
> - Replacing the tables with graphs can make the range problem when the performance gap is significant. Considering this issue, we will replace them properly.
>
> Q2. Nomenclature between body/extractor and head/classifier.
> - We keep the terms "head" and "body" in a revised manuscript.
>
> Q3. Figure 1 modification.
> - We would like to deliver the main message "the clients in FedAvg have different classifiers, however, the clients in FedBABU have the same classifier during local updates" through this figure. Focusing on this main message, we highlighted the main difference by texting as "different classifiers" in FedAvg and "same classifier" in FedBABU, and the definitions of "head" and "body" are explained by a caption. Although "which parts are transferred" is different from FedAvg as you commented, we do not think it needs to be emphasized because it is a natural advantage derived from updating only the body. The personalization step is omitted because this figure delivers the difference during federated training.
>
> Q4. No explicit connection with the "pretrain -> fine-tune" paradigm.
> - "Namely, this problem boils down to how to pre-train a better backbone in a federated learning manner for downstream personal tasks." is inserted in conclusion to motivate FL as the fine-tuning paradigm. We think FedAvg->personalization and FedBABU->personalization follow the pretrain->fine-tune paradigm. It means that developing a single global model via federated training such as FedAvg and FedBABU is a pre-training step. From this perspective, we suggested FedBABU for better representation, which is the goal of pre-training.
>
> Q5. The usage of proper terms: "Unnecessary" or "Irrelevant."
> - We do not consider the change in data distribution. Moreover, we think that "unnecessary" has a negative nuance and "irrelevant" has a neutral nuance. In table 2, training the entire network (i.e., including the head) on the server decreases the personalization performance. Therefore, we used "unnecessary" at least without the change in data distribution.
>
> Q6. Why does the author perform a no-op?
> - We think it is the simplest implementation method if the detailed operation time is not considered. As we reported in Appendix A, training does not require much time even if this implementation method is taken.
>
> Q7. Which part of the analysis/experiments address regularization?
> - Section 5.2.5 addresses FedProx, which regularizes the distance between a global model and local models to prevent local models from deviating during local updates. In this section, we reported "The global models trained by FedProx reduce the personalization capabilities compared by FedAvg" by comparing FedAvg and FedProx.
>
>
> # Typos etc.
> Q1. All of the "i.e."s in the abstract should be removed.
> - We reflected this comment in a revised manuscript.
>
> Q2.  "popular networks have one linear layer...and ResNet" - this sentence is confusing; please revise.
> - We reflected this comment in a revised manuscript.
>
> Q3. Clarify what a "criteria on learning representations" and the "intuition."
> - "The fixed head can be interpreted as the criteria or guideline for each class" is inserted for this sentence. The fixed head can be interpreted as the criteria or guideline for each class because the output (i.e., representation) of the body is pulled toward its label's row of the head through backpropagation. In this vein, we intuitively explain FedBABU as "representation learning based on the same fixed criteria" because FedBABU updates only the body related to representations.
>
> Q4. "When the bottom layers are matched with the body" and "When the top layers are matched with the head" are unclear.
> - We inserted more explanations for the "bottom" and "top" layers by "close to input" and "close to output," respectively. The bottom layers indicate the layers in the forward direction, starting with the input layer. On the contrary, the top layers indicate the layers in the backward direction, starting with the output layer.
>
> Q5. P4 "decayed 0.1 times" --> decayed by a factor of 0.1.
> - We reflected this comment in a revised manuscript.
>
> Q6. Why abbreviate (B), (F) in Table 1? I would suggest "Body Updates" vs. "Head + Body Updates" or similar.
> - We reflected this comment in a revised manuscript.

---

> ### Author Response · Authors · 2021-11-30
> **Additional Response to Reviewer UZKM**
>
> For more empirical evidence related to Table 2 (a motivation for our study), we provide the results on CIFAR10 with the same settings as Table 2. The results show almost the same trend that training the head on the server can hurt personalization, although the performance degradation decreases compared to the case of CIFAR100. It is believed that because the number of classes is smaller and the number of samples per class is larger in the case of CIFAR10, personalization itself is easier, which makes personalization degradation decrease.
>
> |  p  |  Init. (s=10) | Pers. (s=10) |  Init. (s=5)  |  Pers. (s=5) |  Init. (s=2)  |  Pers. (s=2) |
> |:----:|:------------:|:------------:|:------------:|:------------:|:------------:|:------------:|
> | 0.00 | 57.32$\pm$6.76 | 69.11$\pm$5.70 | 46.32$\pm$10.52 | 74.29$\pm$8.29 | 22.29$\pm$16.39 | 85.14$\pm$10.73 |
> | 0.05 (Full) |  64.32$\pm$6.65 | 70.74$\pm$6.01 | 61.03$\pm$8.49 | 73.01$\pm$6.49 | 44.85$\pm$11.44 | 83.22$\pm$7.16 |
> | 0.05 (Body) | 64.52$\pm$5.68 | 75.57$\pm$5.47 | 55.59$\pm$9.82 | 81.15$\pm$6.85 | 38.78$\pm$12.42 | 89.97$\pm$7.05 |
> | 0.10 (Full) | 60.43$\pm$8.35 | 71.05$\pm$6.81 | 49.69$\pm$10.52 | 73.94$\pm$9.11 | 25.80$\pm$17.75 | 84.00$\pm$9.64 |
> | 0.10 (Body) | 59.76$\pm$8.69 | 70.06$\pm$6.36 | 48.47$\pm$8.21 | 76.58$\pm$8.36 | 23.47$\pm$20.46 | 86.53$\pm$11.19 |
>
>
> To ensure generality, we evaluated our proposed algorithm (FedBABU) on various FL settings, datasets, and architectures, as noted by footnote 2 and 3 on page 4. We used MobileNet on CIFAR100 in the main paper. We also used 3ConvNet on EMNIST (Appendix I), 4convNet on CIFAR10 (Appendix F), and ResNet18 and ResNet 50 on CIFAR100 (Appendix G). We designed non-IID situations by sharding the dataset in the main paper and by using Dirichlet distribution (Appendix H). It is believed that our experiments are enough to support our claim (Reviewer 4NJT and yAVA) and exhaustive and comprehensive (Reviewer TnVE and yAVA).
>
> Our paper includes almost all settings that related works have considered. In detail, from the dataset perspective, we used CIFAR10, CIFAR100 and EMNIST for image classification; [a] used CIFAR10, CIFAR100, and FLICKR-AES; [b] used MNIST and CIFAR10; [c] used CIFAR10, CIFAR100 and EMNIST; and [d] used MNIST and CIFAR10. From the architecture perspective, we used 3ConvNet, 4ConvNet, MobileNet, and ResNet; [a] used MobileNet and ResNet; [b] used LeNet-5; [c] used 5-layer CNNs and 2-layer MLP; and [d] used 2-layer MLP. From the basic hyperparameter (i.e., fraction ratio $f$ and local updates $\tau$) of federated learning perspective, we used the combinations of $f \in \\{1.0, 0.1\\}$ and $\tau \in \\{1, 4, 10\\}$. [a] used $f$ of 1.0 and $\tau$ of 4, [b] used $f$ of 0.1 and $\tau$ of 1, and [c] used $f$ of 0.1 and $\tau$ of 1 or 5 depending on the dataset. [d] used $f$ of 0.2 and $\tau \in \\{4, 10\\}$. In summary, our experiments include most of the datasets, architectures, and federated settings in prior works and expand them.
>
> We additionally cited [e] during the rebuttal phase to clarify “training the head” problem, in which the authors mentioned that “debiasing the classifier is promising to directly improve the classification performance.” In a similar context, we proposed FedBABU, which prevents head bias during federated training by stopping updates. In addition, the fact that the performance of FedBABU overwhelms that of FedAvg in various FL settings supports our arguments.
>
> [a] Federated Learning with Personalization Layers (FedPer)
> [b] Think Locally, Act Globally: Federated Learning with Local and Global Representations (LG-FedAvg)
> [c] Exploiting Shared Representations for Personalized Federated Learning (FedRep)
> [d] Personalized Federated Learning: A Meta-Learning Approach (Per-FedAvg)
> [e] No fear of heterogeneity: Classifier calibration for FL with non-IID data

---

### Official Review · Reviewer_yAVA · 2021-10-31

**Correctness:** 3
**Technical Novelty And Significance:** 2
**Empirical Novelty And Significance:** 3
**Recommendation:** 6
**Confidence:** 4

**Main Review:**

Strengths:
- A simple method that retains the FedAvg learning scheme and achieves good results.
- The analyses in the paper are exhaustive and indeed support the author's claim for the most part.
- The paper is written clearly and easy to understand.
- The results seem to be reproducible and code was provided.

Weaknesses:
- Although this is the first paper, that I am aware of, that stresses the importance of learning the feature extractor layers only, I do not consider this idea as an entirely novel contribution of this paper. First, as the authors state, similar ideas are presented in the few-shot learning literature, which is somewhat (yet not entirely) similar in spirit to federated learning setup (hence the success of some FSL methods in FL, such as Per-FedAvg and Reptile). Furthermore, in [1] the authors suggest exactly that learning procedure only with Gaussian processes. I think that this paper should have discussed [1] and compared to it as it also showed large improvements over PFL methods.
- Unless I missed something, I believe that the comparison to baseline methods is not entirely fair. First, it seems that the total number of updates was fixed to 320. Yet, it may be that some baselines require more updates to converge to the optimal solution.
I believe that a better approach to compare with baseline methods would be to set a maximal number of communication rounds and use validation-based early stopping. Then this method could be evaluated against baselines in terms of performance and convergence speed. Second, from the Appendix, it seems that hyper-parameters of baseline methods were not adjusted. This can severely hurt their performance.
- Following the last bullet, did you use a validation split? If not, how did you set the hyper-parameters for your method and baseline methods? Some hyper-parameters, such as the learning rate, may have a detrimental effect on the performance [2], and a good configuration may vary between baseline methods.

General comments:
- Did you train the feature extractor with the nearest template method or in the standard way?
- The current order of the rows in Table 2 makes it a bit hard to compare between F and B rows. I would switch the ordering to the following: p=0,0.05(F),0.05(B),0.1(F),0.1(B).


[1] Achituve, I., Shamsian, A., Navon, A., Chechik, G., & Fetaya, E. (2021). Personalized Federated Learning with Gaussian Processes. arXiv preprint arXiv:2106.15482.
[2] Hsu, T. M. H., Qi, H., & Brown, M. (2019). Measuring the effects of non-identical data distribution for federated visual classification. arXiv preprint arXiv:1909.06335.

**Summary Of The Paper:**

The paper highlights the importance of the shared parameters in personalized federated learning (PFL). The general idea advocated in the paper is that during training, only the feature extractor should be learned and aggregated on the server, while the classifier head should remain fixed. Personalization is achieved after training by fine-tuning the entire local model for each client. Through extensive experiments, the authors support their claim and show improved performance compared to several baseline methods.

**Summary Of The Review:**

Overall it's a nice paper, the results and the analyses are good. Nevertheless, there are some issues. First, it ignores [1] which uses a similar idea. Second, I am not entirely convinced that the empirical evolution of the baseline methods is fair. I am willing to reevaluate the paper based on the author's response and adjustments to the paper.

---

> ### Author Response · Authors · 2021-11-17
> **Response to Reviewer yAVA**
>
> We itemize the weaknesses or comments you mentioned and answer to them.
>
> Q1. Contribution related to the novelty. Recent similar work 'pFedGP.'
> - As we stated in related works, updating partial parameters itself is not completely novel in both meta-learning and federated learning. However, we have highlighted the roles of the body, related to universality for federated training, and the head, related to specialty for personalization.
> - Thank you for pointing out a good reference paper, which is added on page 5 in a revised manuscript. We could not check out the paper you recommended (pFedGP) at the time of submission because the paper was uploaded to arXiv very recently. We have read the paper and found that pFedGP consists of a shared kernel function and a personal GP classifier for personalized federated training. Similar to ours, this work attempts to solve input distribution shifts between clients, focusing on a classifier; they used a GP classifier, while we fixed a linear classifier during federated training. We think that the best advantage of our algorithm is its simplicity.
> - When we, as a reviewer, checked the ICLR 2022 Reviewer Guide, we noticed that the paper should not be judged through comparison with the very recent paper (on or after June 5, 2021). Please excuse that we did not recognize it because it was too recent. If we can, we will add a comparison to this paper.
>
> Q2&Q3. Larger total number of updates. Validation split for early stopping and hyperparameter selection.
> - We fixed the total number of epochs to 320 because the performance gap between algorithms appeared given the fixed epochs. Similarly, In FedRep, the authors fixed the communication rounds to 100 and local epochs to 5 with a client fraction of 0.1 on CIFAR100. We have added the results when the total number of updates is fixed to 640 in Appendix S, agreeing with your opinion. The results show that FedBABU is still the best under considerable heterogeneity, which is the situation federated learning researchers attempt to solve.
> - We did not use a validation dataset for early stopping or optimizing hyper-parameters. Rather, we used the last trained model and followed the hyperparameters based on the provided by [a], which is our codebase, for FedAvg and FedBABU without validation splits. It is thought that hyperparameters applied to FedAvg would be suitable for FedBABU because their learning scheme is similar. And, for other algorithms, we used the proposed hyperparameters in their original papers. We thought that optimizing the hyperparameters for other works was beyond our work. As you mentioned, other methods can hurt their performance in our setting. However, it can imply their weak robustness to the FL settings. Rather, to demonstrate generality, we experimented using various architectures (e.g., 4conv, mobileNet, ResNet), dataset (e.g., CIFAR10, CIFAR100, EMNIST), and FL settings (e.g., various $s$ or Dirichlet, $f$, and $\tau$).
> - A validation-based strategy can be helpful but decreases the number of the training dataset. In particular, each client has only 500 samples in our setting. Therefore, we used training data entirely for training based on hyperparameters in the prior work [a]. Similarly, for MNIST and CIFAR, [a] also used the default hyperparameters provided from “https://github.com/shaoxiongji/federated-learning” for a fair comparison across all baselines, without manually tuning.
> - [a] : Think Locally, Act Globally: Federated Learning with Local and Global Representations
>
> Q4. Training method for feature extractor.
> - We train the feature extractor in a standard (i.e., parametric) way.
>
> Q5. Readability of Table 2.
> - We switched these lines. Please refer to table 2 in a revised manuscript.

---

> > ### Comment · Reviewer_yAVA · 2021-11-26
> > **Response to Authors' comment**
> >
> > I would like to thank the authors for addressing my concerns. Following the authors' comment, I decided to raise the score to 6. In the final version please do add a short discussion on few-shot literature and [1] (which first appeared in arXiv on June 29th). Also, please clarify the hyper-parameter selection strategy and the fact that no validation set was used. This is important information for future studies. Finally, as other reviewers stated the tables are hard to read. A simple solution to that may be to remove the vertical rules and use spaces instead.

---

### Official Review · Reviewer_TnVE · 2021-11-03

**Correctness:** 3
**Technical Novelty And Significance:** 2
**Empirical Novelty And Significance:** 3
**Recommendation:** 6
**Confidence:** 4

**Main Review:**

Pros:
1. The paper takes practical issues of federated learning: what creates degradation problems in training heterogeneous data in federated learning. For me, the problem itself is real and practical.
2. The proposed training scheme is novel and very easy to implement. The authors provide the reason why the model head and the body should be trained separately by comparing the performance (Table 2).
3. This paper provides comprehensive experiments, including both qualitative analysis and quantitative results, to show the effectiveness of the FEDBABU training algorithm in federated learning.

Cons:
1. Although the proposed method provides a study(Table 2), I still suggest the authors provide more investigation of the problem of training the head in federated learning.
How does the randomness of the shared head affect the performance?
Qualitative comparison between training the body only and training the whole network. Maybe plotting the distribution before the head.
How about sharing the fixed random two(or three?) last layers instead of one (head)

pg 5. ‘It is thought that a small s makes local tasks easier because the label distribution owned by each client is limited and the number of samples per class increases.’ <- Could you elaborate this portion more with evidence?



**Summary Of The Paper:**

The paper provides an interesting suggestion in the field of federated learning. In particular, it proposes to enhance federated learning performance by breaking down the network into the body and head and proposes to only train the body during the federated learning phase. Specifically, the authors show that a fixed random classifier can have comparable performance to the learned classifier, and thus propose to share a fixed random classifier across all clients. The authors show the proposed method is efficient especially when the level heterogeneity of the data is more intense.

**Summary Of The Review:**

Overall, I vote for accepting. I like the logical flow that 1) capturing the root of degradation lies in the head, 2) prove the feasibility of using a shared random head, 3) compare the performance between the proposed scheme and the baseline. I think the authors find/tackle the problem in a clear manner. My remaining concern is if it were really the last layer(head) only that creates a degradation problem in a federated learning setting. If so, why? Hopefully, the authors can address my concern in the rebuttal period.

=====POST-REBUTTAL COMMENTS========
Thanks for the authors' response. The author`s responses address well my concern, and I recommend this paper be accepted.

---

> ### Author Response · Authors · 2021-11-17
> **Response to Reviewer TnVE**
>
> We itemize the weaknesses or comments you mentioned and answer to them.
>
> Q1.  More investigation of the problem of training the head in FL.
>
> Q1-1. How does the randomness of the shared head affect the performance?
> - We argued that the orthogonality on the head is necessary to achieve desirable performance in the centralized setting in Section 5.1 and Appendix B. To investigate whether this characteristic is still required under FL settings, we explicitly compare FedBABU with the orthogonal head to FedBABU the orthogonal head.
> - Please refer to Appendix Q in a revised manuscript. The results demonstrate that, as we expected, orthogonality on the head is obviously necessary under FL settings. This is because the non-orthogonal head has a limit to improve performance during local updates.
>
> Q1-2.  Qualitative comparison between training the body only and training the whole network such as plotting the distribution before the head.
> - Please refer to Appendix T in a revised manuscript. We have provided a t-SNE distribution of representations learned by FedAvg (training the whole network) and FedBABU (training the body only) before the head on CIFAR100 and CIFAR10. Furthermore, we describe t-SNE visualization using the sub-classes of one super-class, showing the clustering between similar classes.
>
> Q1-3. Sharing the more random layers instead of one.
> - Please refer to Appendix F.3, where we provided the ablation study according to the fixed parts of the 4conv network on CIFAR10. The results show that FedBABU, which shares only the head, overwhelms other variants.
>
> Q2. `A small $s$ makes local tasks easier’ elaboration.
> - As we explained in Section 3, $s$ is related to the maximum number of classes that each user can have. Shards are constructed mutually exclusively with the same size, and each shard includes samples from the same class. Therefore, small $s$ makes each user have fewer classes but more samples per class. For personalization, therefore, the problem can be more manageable.
> - As an example, let us assume that there are 10 clients, and the cifar10 dataset is scattered to each client. When $s$ is 10, total shards are 100 because 10 (user) $\times$ 10 ($s$). Then, the cifar10 dataset is divided into 100 shards, where each shard consists of 500 samples from the same class. The 100 shards are randomly distributed to the clients so that each client has 10 shards. In this situation, each client can have up to 10 classes. On the contrary, when $s$ is 2, total shards are 20 because 10 (user) $\times$ 2 ($s$). Then, the cifar10 dataset is divided into 20 shards such that each shard consists of 2500 samples from the same class. Here, each client has 2 shards randomly. In this situation, each client can have 2 classes at most, with 2500 samples per class at least. Namely, as $s$ is getting smaller, each client's number of classes is limited (i.e., label distribution owned by each client is limited) and the number of samples per class increases.
> - We have added Appendix R in a revised manuscript for better understanding, where the data distributions of clients according to the shards per user $s$ are provided.
>
> Q3. Does the head only create a degradation problem?
> - Appendix F.3, Section 5.1, and Appendix O can be an answer to this question. In our ablation study (Appendix F.3), the results show that Conv1234(FedBABU) > Conv 123 > Conv12 > Conv1 and Conv1234(FedBABU) > Conv1234+Linear(FedAvg), namely, FedBABU has the best performance. Here, the layer number after Conv or Linear indicates the trained parts during local updates. These results demonstrate that the head only creates a degradation problem.
> - The reason for the former results can be inferred from Section 5.1, which shows that “a randomly initialized fixed body is unacceptable.” With the results of Appendix F.3, it is believed that if the portion of the body is only updated, there is a limit to extracting meaningful representations. Therefore, the entire body should be updated.
> - And, the latter result is explained by Appendix O, which shows that if the head moves even a little bit client by client, then the representation power decreases. Here, we highlight that freezing the head is essential to make all clients have the same class boundary in FL settings.

---

### Official Review · Reviewer_4NJT · 2021-11-03

**Correctness:** 4
**Technical Novelty And Significance:** 3
**Empirical Novelty And Significance:** 3
**Recommendation:** 8
**Confidence:** 4

**Main Review:**

This paper describes enough details and evaluations to understand their proposed methods and its superior performances compared with other methods. The idea itself is simple but there are enough supporting materials in the paper. Supplementary materials also help and  support their proposals. Focusing on the final personalization part and how the whole learning sequence should be is well described.
This paper reveals many advantages against other competitive methods. Among them its robustness with higher performance against data hetegeneity is impressive for practical federated learning. This paper also reveals that  head-part only training is enough for personalization and this leads to faster learning speed. This is also important beneficial performance point in federated learning.
As for weakness the random head part is important in FedBABU and supplemental materials argues this issue but more performance evaluations dependent on head part initialization would support their proposal.

**Summary Of The Paper:**

Federated learning has two development targets of a single global model and multiple personal models under data hetegeneity. This paper focuses on multiple personal model development and proposes FedBABU, Federated Averaging with Body Aggregation and Body Update. In FedBABU they split the whole network into the body part with representation function and the head part with classification function. To make better representation part  is important for personalization and they train the local models and a global mode with the fixed head until personalization. In personalization they train each local model which consists of shared global body part and local specific head part using local data. This paper shows FedBABU outperformers other FedAvg-variant personalization algorithms in many viewpoints. Such as the final performance, robustness against hetegeneity, and faster learning speed.


**Summary Of The Review:**

The problem definition is clear and there are enough materials and descriptions to support their proposals. This paper has supplementary materials and those also support their proposals.
In federated learning data  hetegeneity is a most important issue. As for their performance advantages in comparison with other competitive FedAvg-variant methods, FedBABU is robust against data etegeneity and outperforms others in personalization as well.

---

> ### Author Response · Authors · 2021-11-17
> **Response to Reviewer 4NJT**
>
> We itemize the weaknesses or comments you mentioned and answer to them.
>
> Q1. Performance evaluations dependent on the head part initialization of FedBABU.
> - As you commented, we argued that the orthogonality on the head is necessary to achieve desirable performance in the centralized setting in Section 5.1 and Appendix B. To investigate whether this characteristic is still required under FL settings, we explicitly compare FedBABU with the orthogonal head to FedBABU the orthogonal head.
> - Please refer to Appendix Q in a revised manuscript. The results demonstrate that, as we expected, orthogonality on the head is obviously necessary under FL settings. This is because the non-orthogonal head has a limit to improve performance during local updates.

---

### Author Response · Authors · 2021-11-17
**Overall response**

We thank all the reviewers for their valuable and constructive comments. We have uploaded a revised manuscript, where the added or modified parts are red-colored fonts, and replied to each reviewer.

---

### Public Comment · ~Weiming_Zhuang1 · 2022-02-17
**Relation of FedBABU to Prior Work**

Dear Authors,

Congratulations on your nicely written paper!

I would like to bring your attention to two of my prior works that share similar ideas:
1. Similar to "this problem stems from training the head", we also pointed out that "the last layer captures information that is most related to specific classes" in Section 3.5 of [[1]](https://openaccess.thecvf.com/content/ICCV2021/papers/Zhuang_Collaborative_Unsupervised_Visual_Representation_Learning_From_Decentralized_Data_ICCV_2021_paper.pdf), thus we specially take care of the predictor (i.e. head) update for federated unsupervised learning.
2. Similar to decoupling body and head, we decoupled the backbone & classifier of ResNet and aggregates only backbone for federated person re-identification in [[2]](https://arxiv.org/abs/2008.11560).

Would you mind discussing them in your camera-ready version?

Thank you.

Weiming

[1] Collaborative Unsupervised Visual Representation Learning from Decentralized Data. ICCV 2021.

[2] Performance Optimization for Federated Person Re-identification via Benchmark Analysis. ACMMM 2020

---

> ### Public Comment · ~Jaehoon_Oh1 · 2022-02-18
> **Response**
>
> Dear Weiming Zhuang,
>
> Thank you for your congratulations.
> We will check out your recommended papers and properly cite them in our camera-ready version.
>
> Best, Jaehoon.

---

### Decision · Program_Chairs · 2022-01-20

**Decision:**

Accept (Poster)

**Comment:**

The paper makes some novel and interesting observation pertaining the relationship between data heterogeneity and personalization. Reviewers like the paper and ideas in general but raised several concerns. The rebuttal rectified several confusions and provided more clarification which convinced the reviewers that the paper is above bar for publication.